# Kinetochore mutations and histone phosphorylation pattern changes accompany holo- and macro-monocentromere evolution

Yi-Tzu Kuo [1] ✉, Pavel Neumann [2], Jianyong Chen[1], Jörg Fuchs [1], Veit Schubert [1], Katrin Kumke[1], Mariela Analia Sader[1,3], Michael Melzer[1], Zihao Zhu [1], Axel Himmelbach [1], Heiko Hentrich[4], Jiří Macas [2] & Andreas Houben [1] ✉

Centromeres are essential for kinetochore assembly and spindle attachment. While chromosomes of most species are monocentric with a single centromere, a minority exhibit holocentricity, with a centromere along the chromatid length. Sporadic emergence of holocentricity suggests multiple independent transitions. To explore this, we compare the centromere and (epi)genome organization of two sister genera with contrasting centromere types: *Chamaelirium luteum* with large macro-monocentromeres and *Chionographis japonica* with holocentromeres. Both exhibit chromosome-wide histone phosphorylation patterns distinct from typical monocentric species. Kinetochore analysis reveals similar chimeric *Borealin* in both species, with additional *KNL2* loss and *NSL1* chimerism in *Cha. luteum*. The broad-scale synteny between both genomes supports de novo holocentromere formation in *Chi. japonica*. Despite sharing features with both centromere types, macro-monocentromeres do not represent a direct link between mono- and holo-centromeres. We propose a model for the divergent evolution involving kinetochore gene mutations, altered histone phosphorylation patterns, and centromeric satellite DNA amplification.

Centromeres, the constricted regions of chromosomes that connect sister chromatids, are essential for chromosome segregation in eukaryotes. Most organisms harbor monocentric chromosomes, which are characterized by primary constrictions[1], where the kinetochore assembles and spindle microtubules attach. A minority of species harbor an atypical centromeric organization known as holocentricity, in which centromeres are distributed over the entire chromatid length. Due to the telomere-to-telomere distribution of the holocentromere, sister chromatids cohere throughout their entire lengths and appear in mitotic chromosomes as two parallel structures, without a primary constriction[2]. Each holocentromere comprises multiple 'centromere units', which, depending on the species

[1]Leibniz Institute of Plant Genetics and Crop Plant Research (IPK) Gatersleben, Corrensstrasse 3, Seeland, Germany. [2]Biology Centre, Czech Academy of Sciences, Institute of Plant Molecular Biology, Branišovská 31, České Budějovice, Czech Republic. [3]Multidisciplinary Institute of Plant Biology, National Council for Scientific and Technical Research (CONICET)-National University of Córdoba, Córdoba, Argentina. [4]Molecular Cell Biology, Joseph Gottlieb Kölreuter Institute for Plant Sciences (JKIP), Karlsruhe Institute of Technology, Fritz-Haber-Weg 4, Karlsruhe, Germany. ✉e-mail: kuo@ipk-gatersleben.de; houben@ipk-gatersleben.de

investigated, possess either centromere-specific or non-specific DNA and on which a functional kinetochore protein complex is formed[3]. In contrast, monocentromeres are composed of a single centromere unit per chromosome, although also monocentrics with a few neighboring centromere units per chromosome have been observed, e.g., in *Pisum* and *Lathyrus*[4,5] and the beetle *Tribolium castaneum*[6], forming so-called 'meta-polycentric' chromosomes[5].

The centromere-specific histone H3 (CENH3, also known as CENPA) specifies centromere identity in most species. It serves as a platform for kinetochore assembly, which includes the inner kinetochore constitutive centromere-associated network (CCAN) and the outer kinetochore KMN complex (KNL1c, MIS12c, and NDC80c)[7]. The precise balance between centromere-to-spindle-based tension and sister chromatid cohesion ensures accurate chromosome segregation[8], monitored by the spindle assembly checkpoint (SAC), which verifies proper chromosome-spindle attachments before anaphase. Cell cycle-dependent histone phosphorylation is crucial for recruiting SAC proteins in this process[9].

Histone H3 phosphorylation of the pericentromeric region is part of the histone modification network acting to control centromere function. These modifications can create a permissive environment for the correct assembly and function of centromeres[9]. In plants, with the onset of mitosis, only pericentromeres, where sister chromatids cohere, undergo histone H3 phosphorylation at serine 10 and 28[10,11]. In holocentrics, due to their distinct centromere arrangement, these epigenetic marks are dispersed along the whole length of condensed chromosomes. In monocentric species, meanwhile, both epimarks are enriched adjacent to primary constrictions[11]. Histone H3 threonine 3 phosphorylation originates at pericentromeres in prophase and evenly distributes along chromosome arms at prometaphase in monocentric chromosomes[12,13]. In contrast, histone H2A threonine 120 (H2AT120ph; the threonine position refers to human H2A, and the corresponding positions differ between species) phosphorylation marks mirror the distribution of CENH3 in both centromere types[14,15].

Since holocentric species appear sporadically within phylogenetic lineages that have monocentric chromosomes, it is believed that holocentric chromosomes evolved from monocentric ones. Such a one-way transition happened multiple times in various green algae, protozoans, invertebrates, and different plant families[16]. As a consequence of their independent evolution, holocentromeres are diverse in composition and organization (reviewed in refs. 3,17,18). However, the mechanisms underlying this mono- to holocentromere transition are not yet understood. Analyzing closely related species with contrasting centromere types might illuminate this process. In the parasitic plant genus *Cuscuta*, the mono- to holocentromere transition is associated with extensive changes in genes responsible for the structure and regulation of the kinetochore[19]. In insects such as Lepidoptera, the transition to holocentricity is related to the loss of CENH3[20], leading to a permissive chromatin state-based centromere identity[21] and to CCAN-mediated kinetochore assembly in holocentric taxa[22].

An independent transition to holocentricity occurred in the plant tribe Chionographideae, constituting the holocentric genus *Chionographis* and the monocentric, monotypic sister genus *Chamaelirium*, in the monocot family Melanthiaceae[23–28] (Fig. 1A, Supplementary Fig. 1A, B). The two genera diverged about 23.5 million years ago (mya) and currently exhibit a disjunct distribution, with *Chionographis* found in East Asia and *Chamaelirium* in North America[29]. It has been suggested that the two genera be reclassified as parts of the merged genus *Chamaelirium* due to their otherwise considerable morphological similarities[30].

Notably, the monocentric chromosome-typical primary constriction is absent in *Chamaelirium luteum*[28]. Instead, the species features unusually large heterochromatic centromeres that protrude poleward at metaphase (Supplementary Fig. 1A). It was assumed that the macro-monocentromeres of *Cha. luteum* might represent a precursor state from which holocentromeres in the sister genus *Chionographis* evolved[28]. In the holocentric *Chi. japonica*, each holocentromere is composed of merely 7 to 11, ~1.9 Mb-sized, minisatellite DNA-based, CENH3-positive centromere units that are arranged in a line along the poleward surface of each chromatid[31] (Supplementary Fig. 1B). The centromere units of *Chi. japonica* are up to 200 times larger than those described for the repeat-based holocentromeres in *Rhynchospora* species[32].

To gain insight into the evolution of atypical centromeres accompanying the mono- to holocentromere transition, we resolved the genome and centromere organization of *Cha. luteum*. We compared the (epi)genome and kinetochore composition, as well as the genome synteny, taking advantage of the closely related species *Chi. japonica*, which possesses a contrasting centromere type and the same number of chromosomes ($2n = 24$). Interestingly, the large macro-monocentromeres (up to 15 Mb) of *Cha. luteum* have characteristics in common with mono- and holocentromeres, but, in contrast to initial expectation, they do not constitute a direct link between mono- and holocentromeres. Instead, our comparative analysis of the kinetochore composition revealed a loss of the *KNL2* gene and a chimeric origin of the *NSL1* (a MIS12c component) gene in *Cha. luteum* and of the *Borealin* gene in *Cha. luteum* and *Chi. japonica*. The observed conservation of genome synteny between the two species suggests that the holocentromere formation occurred de novo in *Chionographis*. Additionally, comparison with representatives of the closely related monocentric tribe Heloniadeae revealed a distinct distribution pattern of the cell cycle-dependent histone H3 phosphorylation marks in the tribe Chionographideae. Notably, the distribution of these modifications along the entire length of the chromosomes does not distinguish between holo- and macro-monocentric species. Based on our findings, we discuss possible mechanisms driving the parallel transition from a typical monocentromere to either atypical macro-mono- or even holocentromeres.

## Results

### *Cha. luteum* chromosomes are monocentric despite the lack of a primary constriction

The primary constriction typical for monocentromeric chromosomes is absent in *Cha. luteum*. Instead, large heterochromatic regions protrude on metaphase chromosomes (Supplementary Fig. 1A). To determine whether the so-called macro-centromere [28] acts as an active centromere, a *Cha. luteum*-specific CENH3 antibody was generated as a marker. For this purpose, we analyzed the *Cha. luteum* transcriptome and identified a single *CENH3* gene. Its protein sequence phylogenetically grouped with that of the holocentric *Chi. japonica* (Supplementary Fig. 1D). As expected for CENH3[33], amino acid sequences differed most at the N-terminal tail (Supplementary Fig. 1E). The first twenty N-terminal amino acids were used to generate the anti-CENH3 antibody. Immunostaining of somatic metaphase cells revealed CENH3 signals decorating the protruding macro-monocentromeres (Fig. 1B), where multiple spindle microtubules attach (Fig. 1C, Supplementary Movie 1). During interphase, centromeres formed brightly DAPI-stained, almost equal-sized chromocenters (Fig. 1D). The number of CENH3-positive chromocenters (17–24, on average 20, counted in 30 nuclei) is less than or equal to the number of chromosomes, as in the case of other chromocenter-forming monocentric species, such as *Arabidopsis thaliana*[34]. A preferential distribution of chromocenters close to the nuclear double membrane was revealed (Fig. 1E). A similar nuclear membrane preference of centromeric chromocenters was also found in the holocentric *Chi. japonica* as well as in *A. thaliana*[31]. Our analysis of *Cha. luteum* demonstrates that the chromosome protrusion acts as a centromere and that a primary constriction is not a ubiquitous structural requirement for centromere function in a monocentric species.

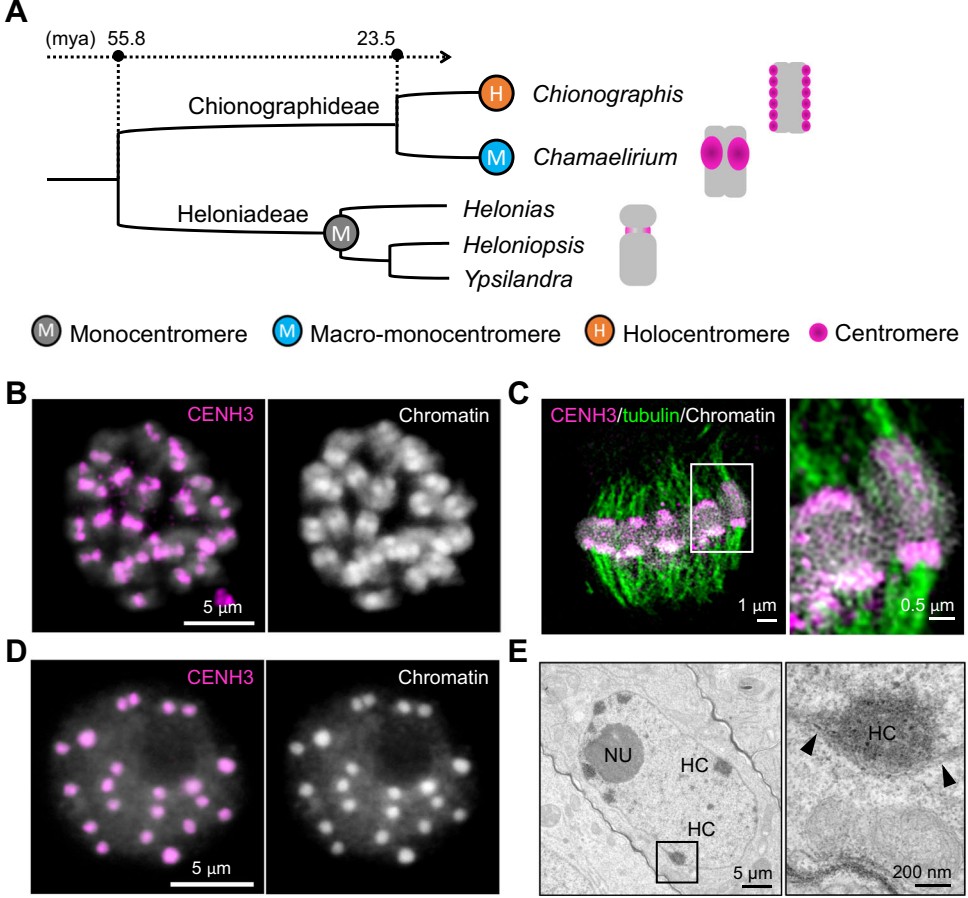

**Fig. 1 | Primary constriction-free *Cha. luteum* chromosomes possess CENH3-positive macro-monocentromeres.** **A** Phylogenetic tree of the tribes Chionographideae and Heloniadeae. Heloniadeae, including the genera *Helonias*, *Heloniopsis*, and *Ypsilandra*, is closely related to Chionographideae, with the two tribes diverging ~ 55 mya[29]. The schemata on the right show the overall chromosome and centromere (magenta) structure. **B** Primary constriction-free metaphase chromosomes of *Cha. luteum* show extensive CENH3 immunosignals (magenta). **C** Microtubules (green) attach to the poleward surface of macro-monocentromeres as observed by super-resolution microscopy (3D-SIM). The enlargement shows the

colocalization between CENH3 (magenta) and microtubules (green). **D** CENH3 signals cluster in brightly DAPI-stained chromocenters of an interphase nucleus. **E** Transmission electron micrograph of a *Cha. luteum* interphase nucleus. **B**–**D** Chromatin was counterstained with DAPI. Electron-dense heterochromatic chromocenters (HC) are often located in proximity to the double-layered nuclear membrane (further enlarged insert, arrows). NU, nucleolus. **B**–**E** At least two independent experiments were carried out to confirm the reproducibility of patterns.

## *Cha. luteum* exhibits exceptionally large satellite-based monocentromeres

The genome size of *Cha. luteum* was determined as 887 Mb/1 C using flow cytometry. To resolve the sequence and size of the *Cha. luteum* macro-monocentromeres, we assembled its genome using a combination of HiFi reads with Hi-C scaffolding, resulting in a ~745 Mb assembly representing ~84% of the *Cha. luteum* genome. The refined assembly retained 92.7% complete BUSCOs across 649 contigs (N50 = 2.1 Mb, L50 = 87) (Supplementary Table 1). Subsequent Hi-C scaffolding generated the top 22 scaffolds, each at least 10 Mb (10−68 Mb) used for downstream analysis (Supplementary Table 2, Supplementary Fig. 2A).

To determine whether the centromeres of *Cha. luteum* are composed of repetitive sequences, we first analyzed the repeat composition of its genome using RepeatExplorer and used the identified repeat clusters as references to assess their enrichment in CENH3-ChIP sequence data. Approximately 49% of the genome is composed of high-copy satellite repeats and transposable elements (Supplementary Table 3). Among these, the 60-bp-monomer satellite cluster CL1, named '*Chama*', is the most abundant repeat (9.3%) (Fig. 2A). Its monomer sequence is rich in centromere-typical dyad symmetries, which may preferentially form a hairpin-loop structure (Fig. 2A). FISH

demonstrated centromeric positioning of the satellite cluster (Fig. 2B), and its enrichment in the CENH3-precipitated fraction is further suggestive of a centromeric nature (Fig. 2C). Additionally, the enrichment of CENH3-ChIP reads in the *Chama* satellite arrays of the assembly (Fig. 2D), as well as the colocalization of CENH3-immuno and *Chama*-FISH signals in interphase chromocenters and heterochromatin regions of metaphase chromosomes (Fig. 2E), confirmed its centromeric localization. To determine the CENH3-positive fraction of the *Chama* arrays, we harnessed super-resolution microscopy. Colocalization of *Chama* FISH and CENH3 immunostaining interphase signals demonstrated that ~60 % of *Chama* arrays interact with CENH3-containing nucleosomes (Supplementary Table 4). Thus, a substantial fraction of the macro-monocentromeric regions forms the active centromere. In addition, the $(AG/CT)_n$-containing CL51 and the GC-rich CL176 repeat cluster, both with very low genome abundance (<0.3%) and no sequence similarity to the *Chama* satellite, were also enriched in the CENH3-precipitated fraction (Fig. 2C).

Among the top 22 scaffolds of *Cha. luteum*, the eight CENH3-interacting *Chama* arrays range in size from 9.58 to 15.30 Mb, with an average size of 11.54 Mb (Supplementary Table 2). Unlike the frequently alternating orientation of the centromeric *Chio* satellite arrays observed in *Chi. japonica*[31], in *Cha. luteum* the sequence arrays are

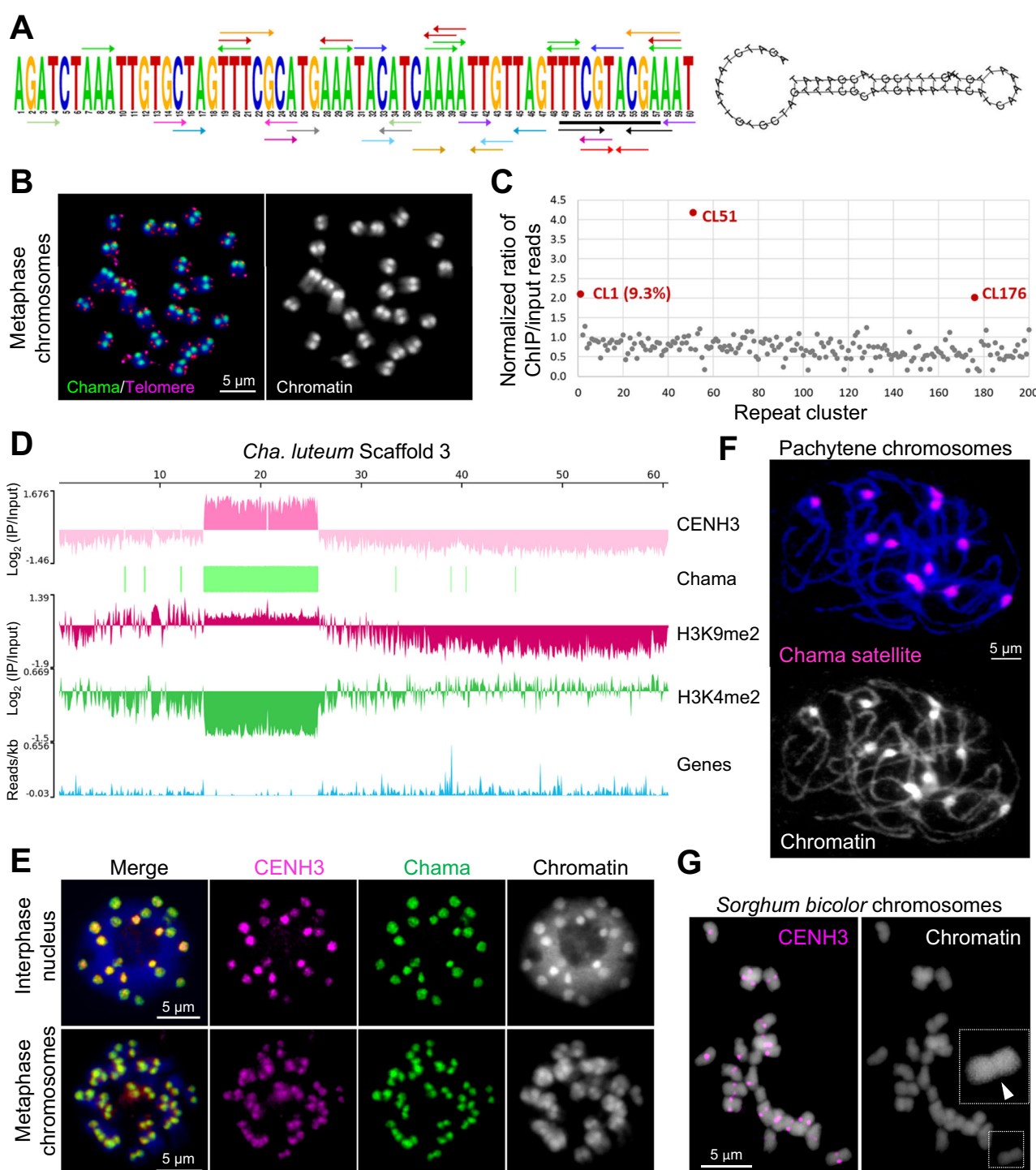

**Fig. 2 | The macro-monocentromeres in *Cha. luteum* are satellite repeat-based.**
**A** The monomer sequence of the *Chama* satellite repeat. The colored arrows indicate dyad symmetries. The 9-bp TTCGTACGA (underlined in black) is shared between the 60-bp *Chama* monomer and the 23-bp *Chio* monomer sequences[31]. Predicted hairpin loop structure formed by a *Chama* monomer. **B** Mitotic metaphase of *Cha. luteum* showing *Chama* (green) and telomere-specific (magenta) signals. **C** The genome proportion and normalized enrichment in CENH3-ChIPseq of the RepeatExplorer clusters. **D** Mapping of the CENH3-, H3K9me2-, and H3K4me2-ChIPseq reads, distribution of *Chama* satellite repeats, and of genic sequences to the 60 Mb-large scaffold 3. Note the strict enrichment of CENH3 on

the *Chama* satellite array. **E** *Cha. luteum* interphase and metaphase chromosomes show colocalization of anti-CENH3 (magenta) and *Chama* repeat-specific (green) immunoFISH signals. **F** *Chama* satellite repeats are located in the knob-like structures of pachytene chromosomes. **G** The immunolabelling of CENH3 (magenta) in monocentric *Sorghum bicolor*. The chromosomal constriction in the enlargement is indicated by an arrowhead. Chromosomes were counterstained with DAPI.
**B**, **E**–**G** At least two independent experiments were carried out to confirm the reproducibility of the labeling patterns. **C** Source data are provided as a Source Data file.

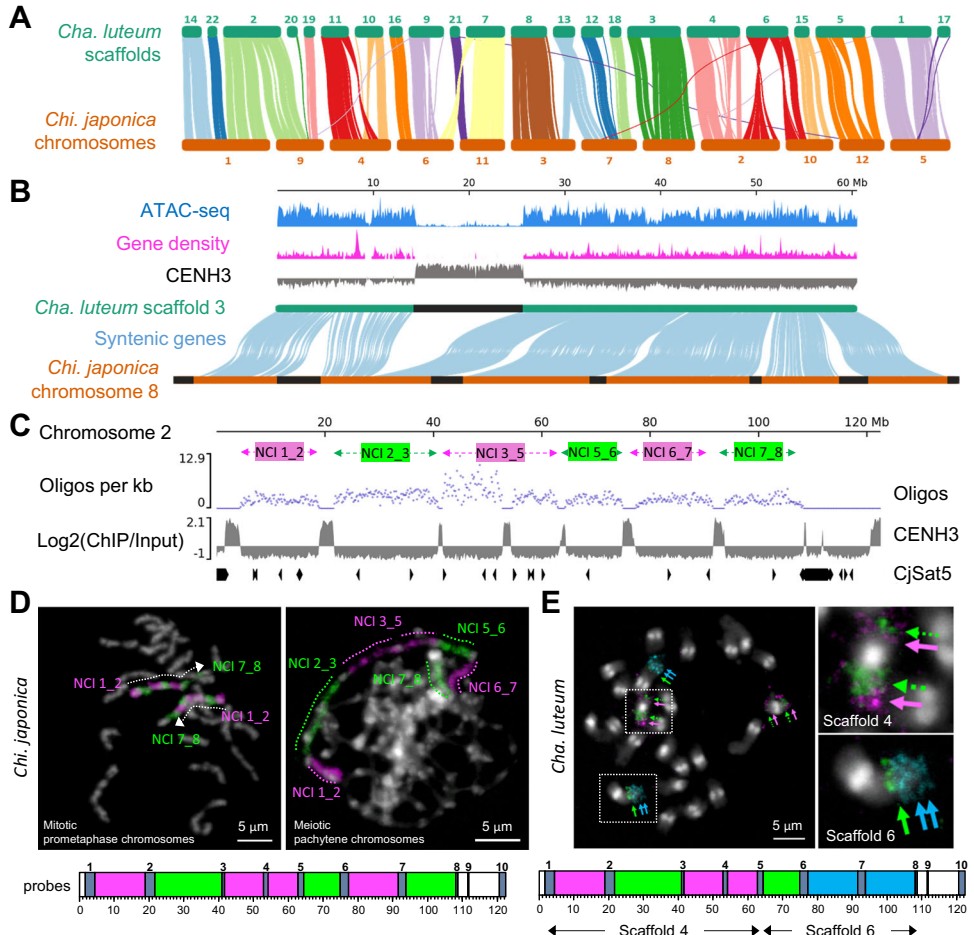

**Fig. 3 | Holocentric *Chi. japonica* and macro-monocentric *Cha. luteum* share broad-scale genome synteny, except for their centromeres. A** Vizualization of syntenic orthologs of coding genes in the assembled scaffolds of *Cha. luteum* and the chromosome-level genome assembly of *Chi. japonica*. **B** Chromosome-level arrangement of orthologs between scaffold 3 of *Cha. luteum* and chromosome 8 of *Chi. japonica* is identical, with the order and orientation of all six non-centromeric intervals of *Chi. japonica* chromosome 8 conserved in scaffold 3 of *Cha. luteum*. The position of the monocentromere in *Cha. luteum* (in black) doesn't correspond to a centromere unit (in black) in *Chi. japonica* according to the syntenic genes. The gene distribution (magenta) and ATAC-seq data (blue) of *Cha. luteum* show that the de novo centromere formation sites in *Chi. japonica* correspond to a high gene density and accessible chromatin state of the corresponding regions in the *Cha.*

*luteum* genome. **C** Design of the oligo-FISH painting probes specific for six non-centromeric intervals (NCIs) between the first and the eighth centromere unit of *Chi. japonica* chromosome 2. Our mapping revealed that the ends of chromosome 2 are highly enriched with the CjSat5 satellite repeat, precluding the design of oligo FISH-painting probes for those specific regions. **D** FISH mapping on mitotic pro-metaphase and meiotic pachytene chromosomes of *Chi. japonica* confirmed the accuracy of the sequence assembly and the specificity of these oligo-FISH probes. **E** In *Cha. luteum*, the *Chi. japonica* chromosome 2 based oligo-FISH signals were located on three arms of two chromosome pairs, corresponding to scaffolds 4 and 6, as predicted by sequence analysis (Supplementary Fig. 4). **D, E** Chromosomes were counterstained with DAPI. At least two independent experiments were carried out to confirm the reproducibility of the labeling patterns.

highly homogeneous and consistently oriented, with higher-order repeat structures maintained across several hundred kilobases and in some cases extending over more than 1 Mb (Supplementary Fig. 2B–D). None of the identified *Chama* arrays was interspersed with retrotransposons. Our FISH analysis of naturally extended pachytene chromosomes, which feature knob-like centromeres, in line with our sequencing results, ruled out the possibility that the macro-monocentromeres of *Cha. luteum* are formed of multiple adjacent centromere units typical for 'meta-polycentric' chromosomes, as the number of *Chama* signals equaled the number of chromosome bivalents (Fig. 2F).

For a direct comparison of the centromere size with a species possessing similar chromosome dimensions, we performed immuno-labeling of *Sorghum bicolor* centromeres with a *S. bicolor*-specific CENH3 antibody, resulting in CENH3 signals at primary constrictions (Fig. 2G)[35]. Despite comparable chromosome sizes (*Cha. luteum* (887 Mb/1 C, $n = 12$) ≈ 73.9 Mb/chromosome, *S. bicolor* (789 Mb/1 C,

$n = 10$) ≈ 78.9 Mb/chromosome), *Cha. luteum* exhibits substantially larger centromeres.

## The genomes of holocentric *Chi. japonica* and macro-monocentric *Cha. luteum* share broad-scale synteny, with the exception of their centromeres

Centromeres play a crucial role in shaping the genome architecture[36]. To investigate whether the evolution of the two centromere types from the sister genera was accompanied by genome reshuffling, we analyzed syntenic orthologs of single-copy coding genes in the assembled scaffolds of *Cha. luteum* and the chromosome-level genome assembly of *Chi. japonica*. We identified 9,960 pairs of collinear genes, revealing large blocks and a high degree of genome conservation (Fig. 3A, Supplementary Fig. 3), although their genomes diverged 23.5 million years ago[29]. In particular, the chromosome-level arrangement of orthologs between chromosome 8 of *Chi. japonica* and scaffold 3 of *Cha. luteum* is identical (Fig. 3B). The order and orientation of

all six non-centromeric intervals of *Chi. japonica* chromosome 8 were conserved in the corresponding chromosome-sized scaffold 3 of *Cha. luteum*. Besides chromosome-sized syntenic regions, 11 large-scale inversions and four inter-chromosomal translocations (scaffolds 2, 6, 7, and 13 of *Cha. luteum*) were found (Fig. 3A, Supplementary Fig. 3). Nevertheless, local gene synteny was mostly conserved. Notably, syntenic blocks were split by centromere units in *Chi. japonica* (Fig. 3B, Supplementary Fig. 3), indicating de novo origin of centromere units in this holocentric species. In addition, ATAC-seq revealed that accessible chromatin regions are evenly distributed across the chromosomes except in the centromeres of *Cha. luteum* (Fig. 3B). This suggests that the de novo holocentromere formation sites *in Chi. japonica* correspond to gene-rich, accessible chromatin regions in the syntenic *Cha. luteum* genome. According to the synteny of centromere flanking regions between the two genomes, the positions of the macro-monocentromeres in *Cha. luteum* (e.g., scaffold 3) and centromere units in *Chi. japonica* mostly do not correspond (Fig. 3B, Supplementary Fig. 3). Assuming that the centromere positions of *Cha. luteum* and of the ancestor of both species are conserved, the loss of monocentromeres was thus likely accompanied by de novo holocentromere formation in *Chionographis*.

To confirm the sequence-deduced conservation of syntenic chromosomal blocks and in silico-identified rearrangements between the two species, we designed oligo-FISH painting probes specific for the six non-centromeric intervals (NCIs) between the first and the eighth centromere unit of *Chi. japonica* chromosome 2 (Fig. 3C). FISH mapping on mitotic prometaphase and meiotic pachytene chromosomes of *Chi. japonica* confirmed the correctness of the sequence assembly and specificity of oligo-FISH probes (Fig. 3D, Supplementary Fig. 4A–C). In *Cha. luteum*, the signals of the six probes were located on three arms of two chromosome pairs (Fig. 3E, Supplementary Fig. 4A–C), corresponding to scaffolds 4 and 6, as predicted by our sequence analysis (Fig. 3A, E, Supplementary Fig. 4D). Additionally, we mapped the sequences of all 12 *Chi. japonica*-based, chromosome-wide oligo pools to the top 22 scaffolds of *Cha. luteum*. This alternative in silico strategy verified the high chromosomal collinearity between the two genomes with contrasting centromere types as well as the absence of chromosome duplications (Supplementary Fig. 5A).

To determine whether this large-scale genome synteny is conserved beyond the tribe Chionographideae, we checked for a potential cross in situ hybridization in the closely related species of tribe Heloniadeae, which includes the genera *Helonias*, *Heloniopsis*, and *Ypsilandra*, and which diverged ~55 mya from tribe Chionographideae (Fig. 1A)[29]. However, the same oligo-FISH painting probes revealed no detectable signals on the chromosomes of *Heloniopsis umbellata*. Moreover, comparative repeat analysis revealed no shared high-copy repeats between *H. umbellata* and *Cha. luteum* or *Chi. japonica* (Supplementary Fig. 5B). Thus, despite their contrasting centromere types, the genomes of *Cha. luteum* and *Chi. japonica* are highly syntenic, yet divergent from those of Heloniadeae species.

**Loss and alteration of kinetochore genes are associated with the transition to unconventional centromeres in Chionographideae**
The protein composition of the kinetochore complex varies considerably across species with different centromere types[18,19,37,38]. To determine whether the evolution of holo- and macro-monocentromeres in different members of the Chionographideae was accompanied by changes of the kinetochore, we investigated the kinetochore protein composition of *Chi. japonica*, *Cha. luteum*, the related monocentric species *Heloniopsis orientalis* from tribe Heloniadeae (Fig. 1A, Supplementary Fig. 1C), and of *Phoenix dactylifera*, a phylogenetically distant monocot species for comparison. In total, we analyzed 29 structural and regulatory kinetochore proteins (Fig. 4, Supplementary Table 5).

In this in silico analysis, we identified homologs for all tested kinetochore proteins in the transcriptomes of *Chi. japonica*, *H. orientalis*, and *P. dactylifera*. In *Cha. luteum*, we found no sequence similar to the CENH3-loading protein KNL2 (Fig. 4A, B). A subsequent BLASTn search for the *KNL2* gene in the genome assembly of this species using the complete gene sequence of *Chi. japonica*[31], revealed only a short fragment at the 3' end of the *KNL2* gene, spanning only two of the seven exons; notably, the exons encoding the conserved SANTA domain and CENPC-k motif of KNL2 were lost (Supplementary Fig. 6A). This *KNL2* fragment in *Cha. luteum* was found in the locus orthologous to the genomic region containing the intact *KNL2* gene in *Chi. japonica* (Supplementary Fig. 6B), indicating that it is a remnant of a formerly intact *KNL2* gene.

Comparative analysis of the identified kinetochore protein sequences revealed a remarkable N-terminal divergence in the MIS12c component NSL1 and the chromosomal passenger complex (CPC) module Borealin. Specifically, the N-terminus of NSL1 in *Cha. luteum* showed unique divergence (Fig. 5A, Supplementary Fig. 7A, B). In contrast, Borealin sequences were similar between *Cha. luteum* and *Chi. japonica*, but the two were distinct from the highly conserved sequences in *H. orientalis* and *P. dactylifera* (Fig. 5B, Supplementary Fig. 7C). Domain analysis revealed that the divergent N-terminus of NSL1 in *Cha. luteum* shares similarity with the BLOC-1 Related Complex Subunit 6 (BORCS6) protein (Supplementary Fig. 7B), while the N-terminus of Borealin in *Cha. luteum* and *Chi. japonica* showed similarity to ribosomal protein S17 (RPS17) (Supplementary Fig. 7C). This suggests a chimeric origin of the *NSL1* gene in *Cha. luteum* and of the *Borealin* genes in both *Cha. luteum* and *Chi. japonica*, with the original N-terminus-coding regions replaced by sequences derived from *BORCS6* and *RPS17* genes, respectively (Fig. 5). By identifying the donors of the N-terminal sequences, we determined the acquired N-terminal lengths to be 104 aa for NSL1 (59% of the protein) (Supplementary Fig. 7B) and 50 aa for Borealin (20% of the protein) (Supplementary Fig. 7C). The chimeric nature of *Borealin* and *NSL1* genes was further confirmed by transcriptional validation and genomic evidence (Supplementary Fig. 8).

NSL1 is a structural component of the MIS12 complex, which consists of NSL1, MIS12, DSN1, and NNF1, and interacts with the other two outer kinetochore complexes (NDC80 and KNL1) via NSL1 and DSN1, as well as with the inner kinetochore proteins via DSN1[39,40]. Therefore, we wondered whether the observed change in NSL1 had an impact on kinetochore assembly in *Cha. luteum*. We took advantage of antibodies developed against the MIS12 and NDC80 proteins of *Chi. japonica* as well as a highly versatile antibody against KNL1 in plants[31,41]. All three antibodies specifically labeled holocentromeres in *C. japonica* and monocentromeres in Heloniadeae species[31,41]. It was anticipated that the antibodies would mark the centromeres of *Cha. luteum* if the kinetochore composition remained unchanged because the sequences of the KNL1, MIS12, and NDC80 antibody target domains are identical between *Chi. japonica* and *Cha. luteum*. We failed to detect signals for KNL1 and NDC80 in *Cha. luteum*, even though we were able to successfully detect MIS12 using well-established immunodetection procedures (Supplementary Fig. 7D). Therefore, we hypothesize that the alteration of NSL1 either prevents KMN complex formation or that these two proteins are present at concentrations below the sensitivity of our immunostaining assay. The presence of the intact MIS12c component DSN1 likely explains why the centromeric recruitment of MIS12 appears to be unaltered.

**Macro-monocentric *Cha. luteum* shows holocentromere-typical cell cycle-dependent histone phosphorylation patterns**
Phosphorylation of histone H3 serves to prime chromatin for faithful chromosome segregation[9]. Thus, we also analyzed the distribution of cell cycle-dependent, spindle assembly checkpoint-associated histone phosphorylation marks. Despite its monocentric chromosomes,

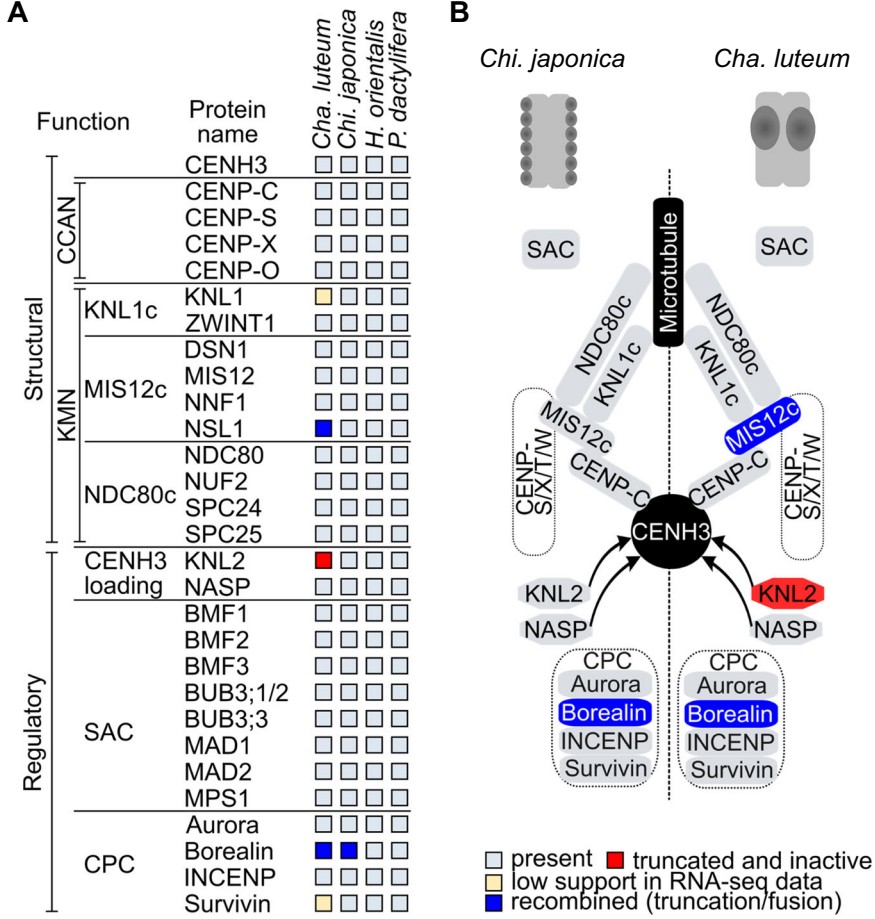

**Fig. 4 | Changes in structural and regulatory kinetochore proteins in Chionographideae. A** Comparative analysis of the kinetochore protein sequences in *Cha. luteum*, *Chi. japonica*, *Heloniopsis orientalis* and *Phoenix dactylifera*. The monocentric species *H. orientalis*, a species from the closely related tribe Heloniadeae, and *P. dactylifera*, a phylogenetically distant monocotyledon, were included for comparison. **B** Simplified schematic illustration of kinetochore structure and centromere organization in the chromosomes of *Chi. japonica* (left) and *Cha. luteum* (right). Proteins or complexes containing proteins that have changed or were lost in *Cha. luteum* or *Chi. japonica* are highlighted in blue or red. Sequence information of the analyzed kinetochore proteins is in Supplementary Table 5. The figure was adapted from Neumann et al.[19].

anti-H3S10/S28/T3 phosphorylation signals were distributed throughout the mitotic metaphase chromosomes of *Cha. luteum* (Fig. 6A), as in holocentric *Chi. japonica*[31] and other holocentric plants[42]. Surprisingly, H2AT120 phosphorylation, an otherwise conserved cell cycle-dependent centromeric mark[15,43], was undetectable in both macro-monocentric *Cha. luteum* and holocentric *Chi. japonica*[31].

To determine whether the holocentromere-like histone phosphorylation patterns evolved before the formation of the tribe Chionographideae, we examined the chromosomal distribution of the phosphorylated histone variants in the sister tribe Heloniadeae (Fig. 1A). Three species, *Helonias bullata*, *Heloniopsis umbellata*, and *Ypsilandra thibetica*, were selected as representatives of the corresponding genera. First, we confirmed the monocentricity of these three species by monitoring the distribution of the conserved outer kinetochore protein KNL1[41] (Fig. 6B, Supplementary Fig. 9). In these species, all analyzed histone H3 phosphorylation patterns were consistent with monocentricity, and H2AT120ph signals were detectable at centromeric regions (Fig. 6B, Supplementary Fig. 9).

To probe the large-scale organization of eu- and heterochromatin in *Cha. luteum*, we assayed the evolutionarily conserved histone marks histone H3K4me2 and H3K9me2. The euchromatin mark H3K4me2 showed uniform signals in metaphase chromosomes, except at centromeres, where the heterochromatin histone mark H3K9me2 was enriched (Fig. 6C). Similar labeling patterns were observed in interphase nuclei with H3K9me2 and H3K4me2 enriched in chromocenters

and euchromatin, respectively (Fig. 6C). These cytological observations were confirmed at the genome-wide level by ChIP-seq, which showed that H3K9me2 is strongly enriched at (peri)centromeric regions (Fig. 2D) and markedly reduced in chromosome arms. In contrast, H3K4me2 is depleted at (peri)centromeric regions and enriched along the chromosome arms, closely resembling the general patterns observed in species with typical monocentromeres.

Thus, although *Cha. luteum* possesses a typical monocentric distribution of eu- and heterochromatin, it exhibits a cell cycle-dependent histone phosphorylation pattern remarkably similar to that of holocentric species, including *Chi. japonica*. This chromosome-wide phosphorylation pattern, unique to Chionographideae species and distinguishing them from Heloniadeae species with typical monocentromeres, likely evolved after the divergence of the two tribes.

## Discussion

### Unraveling centromere evolution in Chionographideae

Our analysis of atypical centromeres provides insights into the evolution of different centromere types. The constriction-free macro-monocentromeres of *Cha. luteum*, a species phylogenetically related to the holocentric *Chi. japonica*, displays features of both mono- and holocentric systems, making them a unique and valuable model for studying centromere evolution. The most parsimonious model consistent with current data suggests that in *Chionographis*, a chromosome-wide spreading of centromere units resulted in the

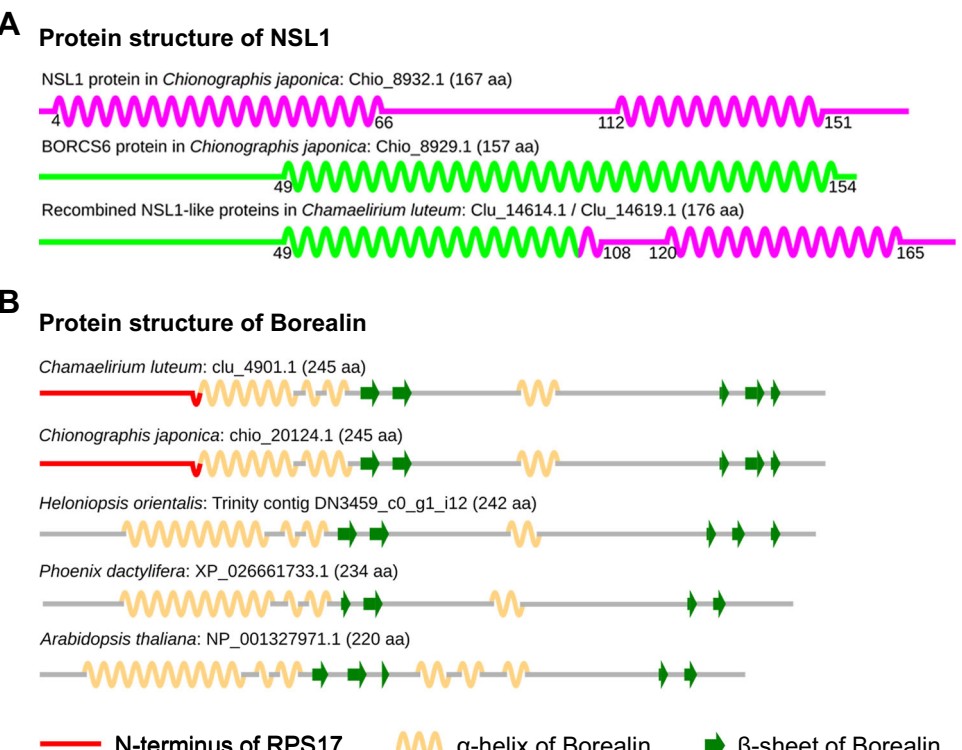

**A  Protein structure of NSL1**

NSL1 protein in *Chionographis japonica*: Chio_8932.1 (167 aa)

BORCS6 protein in *Chionographis japonica*: Chio_8929.1 (157 aa)

Recombined NSL1-like proteins in *Chamaelirium luteum*: Clu_14614.1 / Clu_14619.1 (176 aa)

**B  Protein structure of Borealin**

*Chamaelirium luteum*: clu_4901.1 (245 aa)

*Chionographis japonica*: chio_20124.1 (245 aa)

*Heloniopsis orientalis*: Trinity contig DN3459_c0_g1_i12 (242 aa)

*Phoenix dactylifera*: XP_026661733.1 (234 aa)

*Arabidopsis thaliana*: NP_001327971.1 (220 aa)

— N-terminus of RPS17    〰 α-helix of Borealin    ➡ β-sheet of Borealin

**Fig. 5 | Chimeric origin of the *NSL1* and *Borealin* genes in *Cha. luteum*.**
**A** Comparison of the secondary protein structure of the intact and chimeric NSL1 proteins. NSL1 and BORCS6 proteins are shown in different colors to indicate the origin of the two parts in the chimeric NSL1. **B** Comparison of the secondary structure of Borealin proteins in three Melanthiaceae species, *Phoenix dactylifera*, and *Arabidopsis thaliana*. Although the structure is well conserved in all the species, the first α-helix is significantly shorter in *Cha. luteum* and *Chi. japonica*, which is due to the N-terminus of Borealin being replaced by the N-terminal fragment of ribosomal protein S17 (red). α-helices and β-sheets are shown as beige wavy lines and green arrows, respectively.

formation of a holocentromere (Fig. 7). Given the chromosome-wide synteny and identical chromosome number in both Chionographideae species, it is unlikely that holocentromeres in *Chionographis* arose through multiple fusions of monocentric chromosomes, as has been proposed for the holocentromeres in *Luzula*[44].

In *Cha. luteum* or its unknown ancestor, local centromere expansions likely formed the macro-monocentromeres. A de novo origin of macro-monocentromeres from holocentromeres is unlikely for two main reasons. First, such a process would require multiple fissions of holocentric chromosomes, resulting in a large number of monocentric chromosomes, which is not observed. Second, there are no documented cases of a reversion from holocentricity to monocentricity.

The sister tribes Heloniadeae and Chionographideae, which diverged ~55 mya, exhibit intercontinental disjunctions between eastern Asia and eastern North America[29]. Monocentric species exist in both tribes, implying a monocentric nature of their common ancestors. All *Heloniadeae* species examined possess a primary centromere constriction, suggesting that the loss of this defining chromosome structure occurred after the divergence of the two tribes, but before the evolution of the genera *Chamaelirium* and *Chionographis* at ~23.5 mya (Fig. 7A). While Heloniadeae species display cell cycle-dependent histone phosphorylation patterns typical for monocentricity, *Cha. luteum* and *Chi. japonica*, despite their different centromere structures, share similar distributions of histone H3S10ph, H3S28ph, and H3T3ph, which are typical for holocentricity, coupled with the absence of H2AT120ph. These findings suggest that alterations in (peri)centromeric histone phosphorylation sites also arose after the divergence of the Heloniadeae and Chionographideae lineages.

In addition to the similar histone phosphorylation patterns, both species possess the same mutation in the *Borealin* gene (Fig. 7A-1). The

Borealin protein, a component of the chromosomal passenger complex (CPC), plays a crucial role in regulating histone phosphorylation[45]. The change in the *Borealin* gene might have acted as a molecular trigger, initiating the chromosome-wide distribution of histone H3 phosphorylation and driving the evolution of primary constriction-free chromosomes in Chionographideae. Conversely, constriction-free chromosomes may have arisen first, followed by the selection for the new *Borealin* variant.

The remarkably high proportion (~15%) of centromeric satellite repeats in both species' genomes suggests that the amplification of these repeats was involved in the evolution of both centromere types. Large-scale genome synteny between holocentric *Chi. japonica* and macro-monocentric *Cha. luteum* suggests a de novo origin of holocentromeres in *Chionographis* (Fig. 7A-2). In contrast, local centromere expansion in *Cha. luteum* appears to have driven the formation of macro-monocentromeres (Fig. 7A-3). However, the mechanism underlying the chromosome-wide distribution of de novo-generated centromere units in the holocentric *Chi. japonica* remains unknown[3]. It is likely that this process shares some similarity with neocentromere formation in monocentric species. However, in the case of a neomonocentromeres, the newly formed centromere often forms in the proximity of the "older" centromere[46]. In contrast, the transition from mono-to-holocentromere is accompanied by the formation of numerous de novo centromere units along the entire length of each chromosome. Whether the formation of meta-polycentromeres (e.g. *Pisum*) serve as an intermediate stage in this transition has been debated but not yet proven[2]. Notably, in meta-polycentric *Pisum* species, even centromere units on the same chromosome are based on distinct repeats[4]. By contrast, in *Chi. japonica*, the underlying repeat sequences among centromere units across chromosomes are uniform, indicating a different mode of centromere evolution.

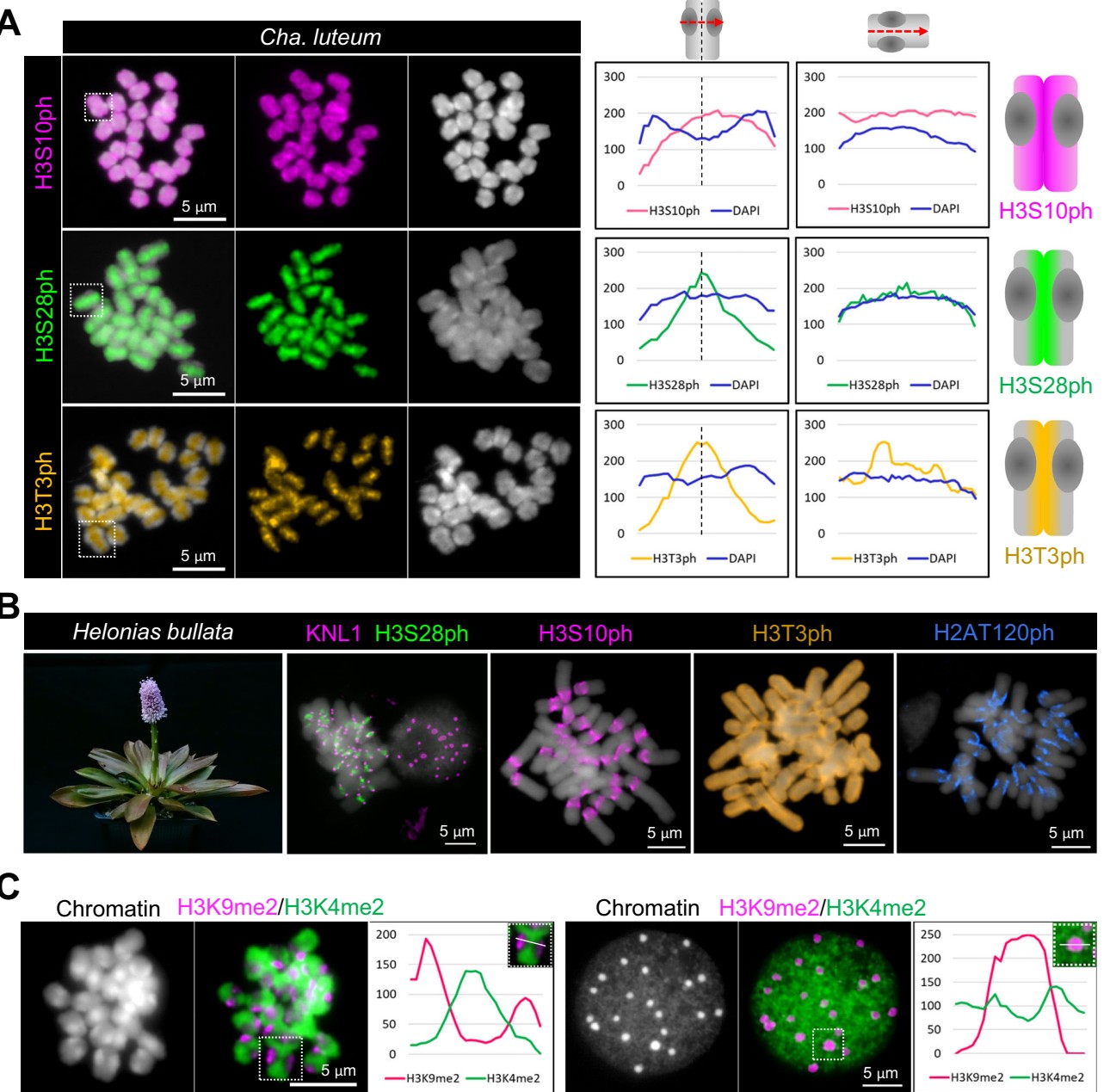

**Fig. 6 | Visualization of cell cycle-dependent and eu- and heterochromatin-specific post-translational histone modifications. A** Mitotic metaphase chromosomes of *Cha. luteum* after immunostaining with antibodies recognizing histone H3S10ph (magenta), H3S28ph (green) and H3T3ph (orange). The line scan plot profiles show the signal intensities of histone marks and DAPI measured in the framed chromosomes (squares). Signal distribution along single chromosomes is depicted as schemata next to the profiles. **B** Mitotic metaphase chromosomes of monocentric *Helonias bullata* after immunostaining with antibodies recognizing a combination of histone H3S28ph (green) and KNL1 (magenta), H3S10ph (magenta), H3T3ph (orange), and H2AT120ph (blue). **C** The immunolabelling patterns of H3K9me2 (magenta) and H3K4me2 (green) on metaphase chromosomes and interphase nuclei of *Cha. luteum* show the large-scale hetero- and euchromatin organization. **A**–**C** Chromosomes were counterstained with DAPI. At least two independent experiments were carried out to confirm the reproducibility of the labeling patterns. Source data are provided as a Source Data file.

Although centromere identity is largely epigenetic, certain sequence motifs or repeat contexts may stabilize CENH3 nucleosome deposition. The centromere repeat monomers in both centromere types share a conserved 9-base pair sequence (TTCGTACGA). This sequence might facilitate the formation of non-B DNA hairpins, a DNA structure known to cause replication fork collapse and subsequent DNA double-strand breaks (DSB)[47]. High DSB rates promote non-allelic (ectopic) recombination repair, which can reposition repeats at new genomic locations even over large distances[48]. Non-B-form DNA-enriched centromeres may represent an ancient form of centromere specification, potentially through interaction with DNA-binding proteins that promote CENH3 loading[49]. It remains unknown whether sequence-driven CENH3 loading occurs in Chionographideae. Further non-allelic gene conversion, a mechanism that accelerates centromere evolution by facilitating sequence exchange among centromere repeats[50,51], might be involved in the mono-to-holocentromere transition process. This mechanism could enable the spreading and homogenization of centromere repeats along the entire length of the chromosomes.

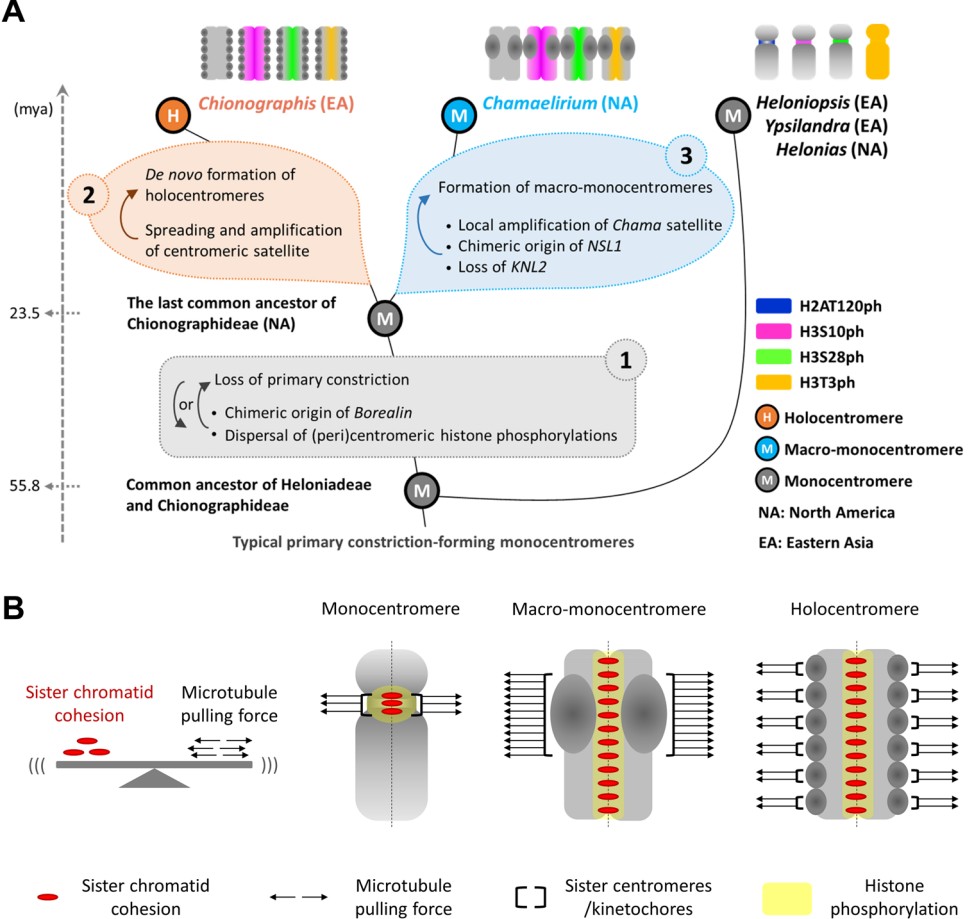

**Fig. 7 | Proposed evolution of holocentromeres and macro-monocentromeres.**
**A** Model explaining the divergent evolution of holo- and macro-monocentric Chionagrahideae. The sister tribes Heloniadeae and Chionographideae, which diverged ~55 million years ago (mya), exhibit intercontinental disjunctions between eastern Asia (EA) and eastern North America (NA). *Chionographis* and *Chamaelirium* possess primary constriction-free chromosomes with holocentromeres and macro-monocentromeres, respectively. In contrast, the monocentromeres of Heloniadeae species possess a primary constriction, suggesting that the loss of this chromosome constriction occurred before the divergence of holocentric *Chionographis* and macro-monocentric *Chamaelirium* ~ 23.5 mya. (1) A Borealin mutation likely acted as an evolutionary trigger, causing the chromosome-wide distribution of histone phosphorylation and sister chromatid cohesion. This ultimately drove the evolution of primary constriction-free chromosomes in the plant tribe Chionographideae. Alternatively, constriction-free chromosomes may have arisen first, followed by selection for the new *Borealin* variant. (2) The evolution of both macro-monocentromeres in *Cha. luteum* and holocentromeres in *Chi. japonica* was driven by the amplification of centromeric repeats. However, in *Chionographis*, this repeat amplification by non-allelic gene conversion was accompanied by the chromosome-wide spread of centromere units, leading to the formation of a holocentromere. (3) In *Cha. luteum*, local centromere expansion resulted in macro-monocentromeres. However, the relationship between loss of *KNL2* and the chimeric origin of *NSL1* and the local centromere expansion remains unclear. **B** Model showing the required balance between sister chromatid cohesion and microtubule pulling forces. Accurate chromosome segregation relies on a balance between sister chromatid cohesion and microtubule-generated pulling forces. As an adaptation required for proper chromosome segregation, the expansion of centromeric regions in both *Chamaelirium* and *Chionographis* probably increased microtubule attachment and pulling forces to counteract the increased sister chromatid cohesion caused by the chromosome-wide distribution of histone H3 phosphorylation. Macro-monocentromeres and holocentromeres represent two distinct outcomes of divergent evolution, each reflecting different adaptations to counteract the increased sister chromatid cohesion. Alternatively, sister chromatid cohesion extended to counteract the increased microtubule pulling forces resulting from the increase in centromere size.

Remarkably, the loss of the *KNL2* gene and the formation of chimeric *NSL1* occurred only in *Cha. luteum*, while both genes remain intact in *Chi. japonica*. Thus, the kinetochore composition of the constriction-free *Cha. luteum* macro-monocentromere appears to diverge more significantly from that of a typical monocentromere than does the holocentromere of *Chi. japonica*. This implies that both centromere types have evolved in parallel after diverging from their last common monocentric ancestor, rather than following a linear transition from macro-monocentric *Chamaelirium* to holocentric *Chionographis*. It remains unknown whether a relationship exists between alterations in *KNL2* and *NSL1* and the local expansion of centromeres in *Cha. luteum*.

### *Cha. luteum* exploits a KNL2-independent mechanism of CENH3 loading and centromere maintenance

Although the mechanism of CENH3 loading in plants is not yet fully understood, it likely depends on KNL2 in most species[52,53]. In *A. thaliana*, which has two KNL2 variants, αKNL2 mutants exhibited reduced levels of CENH3 at centromeres and showed mitotic and meiotic defects but were viable, whereas βKNL2 mutants exhibited complete lethality at the seedling stage[52,53]. The importance of KNL2 for CENH3 loading onto centromeres has also been demonstrated in animals[54–57], suggesting that the role of KNL2 in CENH3 loading is evolutionarily conserved. Melanthiaceae species, like other monocots except grasses[53], possess only a single *KNL2* gene, which encodes a protein structurally more similar to αKNL2 of *A. thaliana*. Despite the loss of

the KNL2 gene, we observed no mitotic defects in Cha. luteum, suggesting that this species exploits an alternative, KNL2-independent mechanism for CENH3 deposition that specifically targets the centromeres. Notably, two holocentric plant species of the genus Cuscuta, C. europaea and C. epithymum, have also been documented to have lost KNL2[19]. In these species, the loss of both αKNL2 and βKNL2 genes was neither lethal nor led to mitotic defects and the authors hypothesized that CENH3 is no longer a centromeric protein, and the two Cuscuta species evolved a CENH3-independent mechanism for attachment to mitotic spindles[19,58]. In Cha. luteum, this is likely not the case, as chromosomes bind to mitotic spindle microtubules exclusively at CENH3-containing domains, suggesting that CENH3 retains its role as a key centromere protein.

Interestingly, all three species lacking KNL2 have unusual centromeres. Centromeric chromatin in Cha. luteum expanded enormously at loci containing highly amplified satellite DNA, comprising up to 15 Mb DNA per centromere. CENH3-containing heterochromatin domains in C. europaea are present at one to three sites per chromosome and are closely associated with the satellite DNA family CUS-TR24, which spans in total 181.2 Mb, corresponding to an average of 25 Mb per chromosome[19]. In contrast, no detectable CENH3 is present on chromosomes in C. epithymum, which contains a low amount of satellite DNA[19]. This shows that the loss of KNL2 has different consequences in different species, possibly dependent on the association of CENH3 with satellite DNA. Conversely, the observation of active KNL2 in the holocentric Chi. japonica suggests that the absence of this gene is not a prerequisite for the evolution of holocentromeres across species.

NASP, a general histone H3 chaperone present in both Chionographideae species, is the only other protein besides KNL2 currently known to participate in CENH3 deposition in plants[59–61]. Although NASP, like KNL2, has been shown to bind to CENH3, the roles of the two proteins differ. While KNL2 ensures CENH3 loading onto centromeres via an interaction with centromeric nucleosomes, NASP does not directly participate in CENH3 deposition. Instead, NASP binds non-nucleosomal CENH3 and escorts it to chromatin assembly factor(s)[59,60]. Thus, it remains unclear which alternative mechanism is responsible for CENH3 loading in Cha. luteum.

### The possible impact of the chimeric Borealin and NSL1 proteins
Chimeric genes are important players in the evolution of genetic novelty[62]. Three ancient domain fusions between kinetochore proteins were predicted to occur in the last common ancestor of eukaryotes[63]. The chimeric origin of NSL1 and Borealin in Chionographideae is unprecedented among kinetochore protein genes.

Borealin is a component of the chromosome passenger complex (CPC), which is a key regulator of mitotic events[64]. The Borealin N-terminus is required for the interaction of CPC with nucleosomes[45]. In Cha. luteum and Chi. japonica, the N-terminus of Borealin has been replaced by a 50 amino acid-long fragment of RPS17. CPC binding to nucleosomes is an upstream requirement for Haspin (phosphorylates H3T3) and Bub1 (phosphorylates H2AT120) activities, and for Haspin/Bub1-mediated CPC enrichment at centromeres. Further, phosphorylated H3T3 is required for centromeric recruitment of the CPC component Aurora B, which phosphorylates H3S10 and H3S28[42,65]. Possibly, a mutation of Borealin that resulted in a chimeric protein led to the observed chromosome-wide distribution of H3S10ph and H3S28ph and the absence of H2AT120ph. Consequently, misregulation of this pathway could disrupt pericentromeric modifications, further affecting chromatid cohesion and chromosome segregation.

NSL1, as a component of MIS12c, is one of the key structural kinetochore proteins. It is essential for interaction with the other two complexes of the outer kinetochore, NDC80c and KNL1c[39]. In Drosophila, NSL1 knockout mutants are lethal, underscoring its essential role for kinetochore function[66]. The effects of the fusion of NSL1 with the N-terminal region of BORCS6 on the formation of the KMN complex in centromeres of Cha. luteum remain unclear. While the lack of KNL1 and NDC80 immunosignals could indicate dramatic changes in kinetochore composition, the results could also arise from antibody incompatibility, conformational masking of epitopes within the kinetochore context, or inaccessibility of antibody target domain. Thus, further investigation is needed to clarify the kinetochore composition and function in Cha. luteum.

BORCS6 belongs to the BLOC-one-related complex (BORC)[67]. In animal cells, BORC controls lysosomal and synaptic vesicle transport and positioning by recruiting ARL8, which either directly interacts with kinesin-3 or indirectly associates with kinesin-1 through cargo adaptors, coupling lysosomes with kinesin motors[67]. Although plant cells do not contain lysosomes, BORC[68], ARL8[69] and kinesin[70] protein homologs are present, and thus BORC might be involved in organelle or vesicle transport as well. It is tempting to speculate that the fusion of BORCS6 N-terminus with the C-terminus of NSL1 could recruit BORC to the kinetochore. If true, chimeric BORC could bring additional function(s) to the kinetochore in Cha. luteum.

Since only 29 structural and regulatory kinetochore genes were considered in our comparative study, we cannot exclude that additional changes in the kinetochore complex were involved in the evolution of holo- and macro-monocentromeres.

### Centromere diversity: different paths to a common functional adaptation
The existence of constriction-free centromeres in Cha. luteum suggests that the primary constriction, typically used to distinguish monocentromeres, is not strictly required for centromere function. Consequently, the absence of this constriction does not necessarily indicate the absence of an active monocentromere. This morphological exception may be explained by the species' immense centromere size and the distribution of histone H3 phosphorylation marks along the entire length of the chromosomes; a pattern otherwise characteristic of holocentromeres.

Accurate chromosome segregation relies on a delicate balance between sister chromatid cohesion and microtubule-generated pulling forces[71] (Fig. 7B). Consequently, as an adaptation required for proper chromosome segregation, the expansion of centromeric regions in both Chamaelirium and Chionographis probably increased microtubule attachment and pulling forces to offset the increased sister chromatid cohesion brought on by the chromosome-wide distribution of pericentromeric histone H3 phosphorylation. This sequence of events is more likely than the alternative scenario of independent histone H3 phosphorylation expansions in both genera, followed by centromere expansion. Macro-monocentromeres and holocentromeres represent distinct outcomes of divergent evolution and different adaptations to counteract the increased sister chromatid cohesion. These divergent solutions highlight the plasticity of centromere evolution in response to selective pressure imposed by changes in chromatid cohesion dynamics driven by mutation in a single gene (chimeric Borealin). Moreover, these findings suggest a complex interplay between centromere size, number and distribution of centromere units (mono versus holo), and chromatid cohesion in shaping chromosome morphology. Overall, the two Chionographideae species offer valuable insights into centromere type evolution, highlighting the importance of incorporating greater phylogenetic diversity in model organisms used to study fundamental chromosomal features.

In summary, we suggest that the divergent evolution of both atypical centromeres is the result of a complex, stepwise interplay involving kinetochore gene mutations, alterations of mitotic histone phosphorylation patterns, and amplification of centromeric satellite DNA.

## Methods

### Plant materials

*Chamaelirium luteum* (L.) A. Gray plants used in this study were provided by the Deutsche Homöopathie-Union (DHU), Germany, *Chionographis japonica* (Willd.) Maxim. plants were obtained from commercial nurseries in Japan, and the *Helonias bullata* L., *Heloniopsis umbellata* Baker, *Heloniopsis orientalis* var. *breviscapa*, and *Ypsilandra thibetica* Franch. species were purchased from British nurseries in the UK. The plants were grown at IPK Gatersleben (Germany) in a greenhouse: 16 h light (from 6 AM to 10 PM), day temperature 16 °C, night temperature 12 °C. Seeds of *Sorghum bicolor* BTx623 and *Secale cereale* L. inbred line Lo7 were obtained from the IPK GeneBank (Gatersleben, Germany) and were germinated on wet filter papers to harvest roots and young leaves for experiments.

### Genome size measurement

To isolate nuclei, approximately 0.5 cm² of fresh leaf tissue of *Cha. luteum* was chopped together with equivalent amounts of leaf tissue of either of the two internal reference standards *Glycine max* (L.) Merr. convar. max var. max, cultivar 'Cina 5202' (Gatersleben genebank accession number: SOJA 392; 2.21 pg/2 C) or *Raphanus sativus* L. convar. sativus, cultivar 'Voran' (Gatersleben genebank accession number: RA 34; 1.11 pg/2 C), in a petri dish using the reagent kit 'CyStain PI Absolute P' (Sysmex-Partec) following the manufacturer's instructions. The resulting nuclei suspension was filtered through a 50-µm CellTrics filter (Sysmex-Partec) and measured on a CyFlow Space flow cytometer (Sysmex-Partec, Germany). The gating strategy is shown in Supplementary Fig. 10. At least six independent measurements were performed for *Cha. luteum*. The absolute DNA content (pg/2C) was calculated based on the values of the G1 peak means and the corresponding genome size (Mbp/1C), according to ref. 72.

### Short-read sequencing of DNA and RNA

Genomic DNA of *Cha. luteum* was extracted from leaf tissue using the DNeasy Plant Mini kit (Qiagen, Germany). Low-pass paired-end (2×150 bp) genome sequencing was performed using DNBSEQ system by BGI (China). Total RNAs from leaf, root, and fruit tissues of *Cha. luteum* were isolated using the Spectrum™ Plant total RNA kit (Sigma, USA, cat. no. STRN50). Library preparation (Illumina Stranded mRNA Prep Ligation Kit) and sequencing at IPK Gatersleben or Novogene (UK) (paired-end, 2 × 151 cycles, Illumina NovaSeq6000 system) involved standard protocols from the manufacturer (Illumina Inc., USA).

### Isolation of HMW DNA, HiFi library preparation, and sequencing

For long-read PacBio sequencing, high-molecular weight (HMW) DNA of *Cha. luteum* was isolated from leaves of a single plant using the NucleoBond HMW DNA kit (Macherey Nagel, Germany). Quality was assessed using the FEMTO Pulse system (Agilent Technologies Inc., CA, USA). Quantification involved the Qubit device and the dsDNA High Sensitivity assay kit (Thermo Fisher Scientific, MA, USA). A HiFi library was prepared from 15 µg HMW DNA using the "SMRTbell prep Kit 3.0" according to the manufacturer's protocol (Pacific Biosciences of California Inc., CA, USA). The initial DNA fragmentation was performed using the Megaruptor 3 device (Shear speed: 29; Diagenode, Belgium). Finally, HiFi libraries were size-selected (narrow-size range: approximately 20 kb) using the SageELF system with a 0.75% Agarose Gel Cassette as described by the manufacturer (Sage Science Inc., MA, USA). Sequencing (HiFi CCS) was performed using the Pacific Biosciences Revio device (24 h movie time, 155 pM loading concentration, 2 h pre-extension time, diffusion loading, 100 min loading time, 22 kb mean insert length according to SMRT link raw data report, 55 Gb HiFi CCS yield) following standard manufacturer's protocols (Pacific Biosciences of California Inc., Menlo Park, CA, USA) at IPK Gatersleben.

### Chromosome conformation capture (Hi-C) sequencing

Hi-C sequencing libraries were generated from flowers of *Cha. luteum* as described previously[73] using *Dpn*II enzyme, and were sequenced (paired-end, 2 × 111 cycles) using the NovaSeq6000 device (Illumina Inc., USA) at IPK Gatersleben. A total of ~102 Gb paired-end reads were generated. After filtering, ~87 Gb Hi-C read pairs were used for scaffolding.

### Genome assembly and Hi-C scaffolding

A total of ~53 Gb of PacBio HiFi reads (~59.5× coverage) of *Cha. luteum* were assembled into contigs using hifiasm (v0.19.3-r572; default)[74]. Contig statistics were calculated with Quast (v2.3)[75], and gene content completeness was evaluated with Benchmarking Universal Single-Copy Orthologs (BUSCO) (v4.1.2; dataset: liliopsida_odb10, E-value = 0.001)[76]. An initial 1.56-Gb primary assembly of *Cha. luteum* genome was constructed with a BUSCO completeness of 95.6% (Supplementary Table 1). The unexpectedly large assembly size and the high score of duplicated BUSCOs (68.0%) suggested the presence of notable heterozygosity in the primary contigs. To join the residual heterozygous contains, purge_dups (v 1.2.5; default)[77] was used to identify and remove duplicated sequence segments in the primary assembly, resulting in a reduction of duplicated BUSCOs to 10.6% (Supplementary Table 1). The Arima Genomics mapping pipeline (https://github.com/ArimaGenomics/mapping_pipeline) was used to process the Hi-C data, including read mapping to the contigs, read filtering, read pairing, and PCR duplicate removal, and scaffolding was performed using YaHS (v1.2a.2; -e GATC --no-contig-ec)[78]. Hi-C contact maps and manual curation were accomplished by the bash scripts provided (https://github.com/c-zhou/yahs) and visualized using Juicebox (https://github.com/aidenlab/Juicebox).

### Gene-based synteny analysis

The genome of *Chi. japonica* was re-annotated by mapping the clean RNA-seq data (EMBL ENA PRJEB58123) generated from our previous study[31]. For genome-directed transcriptome assembly, the RNA-seq datasets generated in this study from fruits (~7 Gb), leaves (~13 Gb), and roots (~13 Gb) of *Cha. luteum* (EMBL ENA PRJEB82608), were aligned to the assembled scaffolds using HISAT2 (v2.2.1, default parameters)[79], and then processed to produce gene feature annotation with StringTie (v2.1.1, default parameters)[80]. Two sets of non-redundant transcripts from *Chi. japonica* and *Cha. luteum* were generated using gffread (v0.12.6)[81]. TransDecoder (v5.5.0)[82] was used to annotate coding regions in transcripts. BRAKER3 pipeline was used to improve the annotation accuracy of protein-coding genes[83].

To find the links of conserved single-copy proteins between the *Chi. japonica* and *Cha. luteum* genomes, a Python script was developed as follows. First, the translated protein sequences of *Chi. japonica* were aligned to those of *Cha. luteum* via blastp (v2.5.0, default). Second, alignments with an identity of <90% and an alignment length of <150 aa were treated as noisy alignments and were filtered out. Third, only when a sequence showed similarity to a single gene in the other genome were considered as conserved single-copy proteins. Finally, the genomic positions of the syntenic single-copy proteins were extracted from the gff3 files to create the links between the two genomes. NGenomeSyn (v1.41)[84] was used to visualize the genome synteny.

### Transcriptome-based identification of kinetochore protein genes

The clean RNA-seq datasets from various tissues of *Cha. luteum* were de novo assembled using Trinity 2.4.0[82,85] with default parameters. In addition, the genome-directed assembled transcriptome of *Cha. luteum* as described above was also applied for downstream analyses. Putative coding regions were first identified by TransDecoder (v5.5.0)[82] with a minimum protein length of 100 as a threshold.

Trinity-made transcriptome assembly of *Chi. japonica* was obtained from our previous study[31]. The RNA-seq data and genome assembly used in the study were applied here for genome-directed transcriptome assembly using HISAT2 (v2.2.1, default)[79] followed by StringTie (v2.1.1, default)[80], as described above. De novo transcriptome assembly in *H. orientalis* was constructed using RNA-seq data downloaded from Sequence Read Archive (https://www.ncbi.nlm.nih.gov/sra; SRR28160651 and SRR28160654).

Protein sequence databases for these species were constructed by translating predicted open reading frames from de novo and genome-directed transcriptome assemblies generated using Trinity (all three species) and StringTie (*Chi. japonica* and *Cha. luteum)*, respectively. The analysis was done using two types of sequence data; gene annotation models predicted in genome assemblies based on RNA-seq data and transcriptome assemblies.

Kinetochore protein sequences were first identified in *P. dactylifera* using BLASTp searches in the GenBank protein database, guided by sequences from previous studies[19,37,86,87]. These identified sequences were then used to search for homologs in the three Melanthiaceae species. The sequences of kinetochore proteins translated from de novo and genome-derived transcriptome assemblies matched well. Since the genome-derived assemblies were coupled with detailed information about the gene structures, we used these for further analyses. In a few cases, the gene structures had to be manually corrected by reassembling the RNA-seq reads assigned to specific loci and using the programs est2genome[88] and genewise[89] for the gene annotations.

## Reverse transcription polymerase chain reaction (RT-PCR)

Total RNA was extracted from roots and young leaves of *Cha. luteum* using the Spectrum™ Plant Total RNA-Kit (Sigma-Aldrich). RNA samples were mixed and treated with DNase using the on-column DNA removal protocol (RNase-Free DNase I Kit, Norgen Biotek). cDNA was synthesized with RevertAid First Strand cDNA Synthesis Kit (Thermo Scientific) using oligo(dT) primer. PCR amplification was performed using gene-specific primers (Clu_NSL1_F1:5′- CGACGGGAAATGGGAGG-3′; Clu_NSL1_R1:5′- CAAATAAAATTTCGAACAAGTCTGCT-3′; Clu_Borealin_F1:5′-GGGTATTCCTCAGCTCAAAGA-3′; Clu_Borealin_F1:5′-GCCTCCAAGTTATCGTCCTT-3′, Tm = 55 °C) with GoTaq DNA Polymerase (Promega) following the manufacturer's instructions. PCR products were size-verified on 1 % agarose gels, and purified amplicons were Sanger sequenced by Eurofins (Germany).

## Phylogenetic analysis

The CENH3 protein sequences of *Cha. luteum* and the other species derived from the NCBI GenBank (Supplementary Table 6) were aligned using the ClustalW algorithm implemented in MEGA X by default setting[90,91]. The maximum-likelihood tree was constructed via IQ-Tree web server (http://iqtree.cibiv.univie.ac.at/)[92] and visualized using Interactive Tree Of Life (iTOL, http://itol.embl.de/)[93,94].

## Antibody production

The synthesized peptides of *Cha. luteum* CENH3 (ClCENH3: MAPTKKTKKTTENINNRPAL-C) were used for the immunization of rabbits to generate polyclonal antibodies. The peptide synthesis, immunization, and antibody purification were performed by LifeTein (www.lifetein.com, USA).

## Indirect immunostaining

Mitotic chromosomes and interphase nuclei were prepared from root meristems. Roots were pretreated in ice-cold water overnight and fixed in 4% paraformaldehyde in Tris buffer (10 mM Tris, 10 mM EDTA, 100 mM NaCl, 0.1% Triton X-100, pH 7.5) for 5 min on ice under vacuum treatment, followed by another 25–30 min solely on ice. Root meristems were then chopped in lysis buffer LB01 (15 mM Tris, 2 mM

Na$_2$EDTA, 0.5 mM spermine, 80 mM KCl, 20 mM NaCl, 15 mM β-mercaptoethanol, and 0.1 % (v/v) Triton X-100)[95], the cell suspension was filtered through a 50-μm CellTrics filter (Sysmex-Partec) and subsequently centrifuged onto slides using a Cytospin3 (Shandon, Germany) at 700 rpm (×55.32 g) for 5 min. The chromosome spreads were blocked in 3% BSA in 1× phosphate-buffered saline (PBS) at room temperature (RT) for 1 h and incubated with primary antibodies in 1% BSA/ 1× PBS at 4 °C overnight. After three washes in 1× PBS at RT for 5 min each, secondary antibodies in 1% BSA/ 1× PBS were applied, followed by an incubation at 37 °C for 1 h. After three washes, the slides were dehydrated in 70-90-100% ethanol series for 3 min each and counterstained with 10 μg/ml 4′,6-diamidino-2-phenylindoline (DAPI) in Vectashield antifade medium (Vector Laboratories, USA). For immunodetection of microtubules, root pretreatment with ice-cold water was omitted, and the Tris buffer and 1× PBS mentioned above were substituted by 1× MTSB buffer (50 mM PIPES, 5 mM MgSO$_4$, and 5 mM EGTA, pH 7.2).

The primary antibodies used in this study included customized rabbit anti-*Cha. luteum* CENH3 (dilution 1:500), rabbit anti-*Chi. japonica* MIS12 (dilution 1:100)[31], rabbit anti-*Chi. japonica* NDC80 (dilution 1:100)[31], and rabbit anti-*Cuscuta europaea* KNL1 (dilution 1:500 or 1:1000)[19], as well as the commercially available mouse anti-alpha-tubulin (Sigma-Aldrich, USA, cat. no. T9026-2, dilution 1:300), rabbit anti-histone H3K4me2 (abcam, UK, cat. no. ab7766, dilution 1:300), mouse anti-histone H3K9me2 (abcam, UK, cat. no. ab1220, dilution 1:300), mouse anti-histone H3S10ph (abcam, UK, cat. no. ab14955, dilution 1:1000), rat anti-histone H3S28ph (Sigma-Aldrich, USA, cat. no. H9908, dilution 1:1000), rabbit anti-H3T3ph (Sigma-Aldrich, USA, cat. no. 07-424, dilution 1:1000), and rabbit anti-H2AT120ph (Active Motif, USA, cat. no. 61196, dilution 1:500).

The anti-rabbit rhodamine (Jackson ImmunoResearch, USA, cat. no. 111-295-144, dilution 1:300), anti-rabbit Alexa488 (Jackson ImmunoResearch, USA, cat. no. 711-545-152, dilution 1:300), anti-mouse Alexa488 (Jackson ImmunoResearch, USA, cat. no. 715-546-151, dilution 1:300), and anti-rat Alexa488 (Jackson ImmunoResearch, USA, cat. no. 112-545-167, dilution 1:300) were used as secondary antibodies.

## Repeatome analysis

The low-coverage genome skimming dataset of *Cha. luteum* was generated in this study, and those of *Chi. japonica* (ERR10639507, EMBL ENA, https://www.ebi.ac.uk/ena/)[31] and *H. umbellata* (SRR15208642, NCBI SRA, https://www.ncbi.nlm.nih.gov/sra/)[96] were publicly available. Genomic PE reads were assessed by FastQC[97] implanted in the RepeatExplorer pipeline (https://repeatexplorer-elixir.cerit-sc.cz/galaxy/) and filtered by quality with 95% of bases equal to or above the cut-off value of 10. Qualified PE reads of *Cha. luteum* equivalent to 0.5× genome coverage were applied to analyze repetitive elements by a graph-based clustering method using RepeatExplorer[98–100]. The automatic annotation of repeat clusters was manually inspected and revised if necessary, followed by a recalculation of the genome proportion of each repeat type. The comparative clustering analysis was performed based on one million PE reads from each of the three species.

## Chromatin immunoprecipitation (ChIP) sequencing

The CENH3-ChIP experiment was performed with minor modifications as described by Kuo et al.[31]. 0.65 g of *Cha. luteum* flower and 1.0 g of *Secale cereale* (inbred line Lo7) leaf tissue were ground with liquid nitrogen and homogenized separately in 10 ml nuclei isolation buffer (1 M sucrose, 5 mM KCl, 5 mM MgCl$_2$, 60 mM HEPES pH 8.0, 5 mM EDTA, 0.6% Triton X-100, 0.4 mM PMSF, 1 μM pepstatin A, cOmplete protease inhibitor cocktail (Roche)). Nuclei fixation was performed in 1% PFA/ nuclei isolation buffer at RT, 12 rpm for 10 min and terminated by adding glycine to a final concentration of 130 mM. The nuclei suspension was filtered through Miracloth (Millipore) twice and a 50-μm CellTrics filter (Sysmex) once and centrifuged at 4 °C, 3,000 ×g for

10 min. The nuclei pellet was resuspended in 1 ml extraction buffer (0.25 M sucrose, 10 mM Tris-HCl pH 8.0, 10 mM MgCl₂, 1% Triton X-100, 1 mM EDTA, 5 mM β-mercaptoethanol, 0.1 mM PMSF, 1 μM pepstatin A, cOmplete protease inhibitor cocktail), followed by centrifugation at 4 °C, 12,000$g$ for 10 min. After removing the supernatant, nuclei were resuspended in 150 μl of nuclei lysis buffer (20 mM Tris-HCl pH 8.0, 10 mM EDTA, 1% SDS, 0.1 mM PMSF, 1 μM pepstatin A, cOmplete protease inhibitor cocktail). Chromatins were sonicated for 14 cycles of 30 s ON, 30 s OFF at high power in a Bioruptor (Diagenode), followed by an addition of 100 μl ChIP dilution buffer (16.7 mM Tris-HCl pH 8.0, 167 mM NaCl, 1.1% Triton X-100, 1 mM EDTA, cOmplete protease inhibitor cocktail), and continued sonication to a total of 31 cycles under the same setting. The sonicated samples were diluted 10 times with ChIP dilution buffer, centrifuged at 4 °C, 13,000$g$ for 5 min, and the supernatant of each sample was transferred to new tubes. To dilute the high proportion of the putative *Cha. luteum* centromeric repeat, sonicated chromatin *of Cha. luteum* and *S. cereale* were mixed in a 1:3 ratio. The mixed chromatins were incubated with the ClCENH3 antibody (10 mg/ml) to a final 1:500 dilution at 4 °C by shaking at 14 rpm for 12 h. Dynabeads™ Protein A (Invitrogen) in ChIP dilution buffer, corresponding to 0.1× volume of the chromatin solution, was added to the antibody-prebound chromatins and incubated at 4 °C by shaking at 14 rpm for 1.5 h. The collected beads were then washed twice in low salt buffer (150 mM NaCl, 0.1% SDS, 1% Triton X-100, 2 mM EDTA, 20 mM Tris-HCl pH 8.0), followed by three washes in high salt buffer (500 mM NaCl, 0.1% SDS, 1% Triton X-100, 2 mM EDTA, 20 mM Tris-HCl pH 8.0), and another two washes in TE buffer at 4 °C by shaking at 14 rpm for 5 min. The bead-bound chromatin was purified by using iPure kit v2 (Diagenode) following the manual and quantified using Qubit™ dsDNA HS Assay kit (Invitrogen). ChIPseq libraries were prepared by NEBNEXT® Ultra™ II DNA Library Prep Kit for Illumina (New England Biolabs) and sequenced using NovaSeq 6000 system (Illumina) by Novogene (UK) in the paired-end run (2×150 bp).

For histone mark ChIP, chromatin was isolated from 0.33 g of *Cha. luteum* flowers following the same nuclei isolation and sonication procedure described above. Antibodies H3K9me2 (ab1220, abcam) and H3K4me2 (ab7766, abcam) were added to a final dilution of 1:100. The subsequent immunoprecipitation, washing, DNA purification, and library preparation steps were performed as described for the CENH3-ChIP. The sequencing using DNBSEQ system was performed by BGI (China).

## ChIP-seq data analysis

To evaluate the enrichment of repeats associated with CENH3-containing nucleosomes, single-end reads of CENH3-ChIP-seq and input-seq were quality filtered using the tool "Processing of FASTQ reads" (Galaxy Version 1.0.0.3), implemented in the Galaxy-based RepeatExplorer portal (https://repeatexplorer-elixir.cerit-sc.cz/galaxy/). ChIP-Seq Mapper (Galaxy version 1.1.1.4) (Neumann et al.[5]) was used to map the ChIP- and input-seq reads on RepeatExplorer-derived contig sequences of repeat clusters. To analyze the size and position of *Cha. luteum* centromeres and histone modification patterns, the paired-end reads of ChIP- and input-seq were quality-filtered by Trimmmomatic (Galaxy Version 0.39)[101] and the resulting reads were mapped to the *Cha. luteum* genome assembly using Bowtie2 (Galaxy Version 2.5.3)[102] with default parameters. No filtering to remove multimapping reads was applied; therefore, all analyses include both uniquely and multimapping reads. Peak calling (--broad −887400000) was performed with MACS (v3.0.0)[103]. The deeptools bamCompare function (Galaxy Version 3.5.4)[104] was used to generate the normalized ChIP-seq signal track as the average log2-ratio of ChIP over input read counts in genome-wide 1 kb windows. Visualization of chromosome regions with multiple tracks were plotted with pyGenomeTracks (Galaxy version 3.8)[105].

## ATAC sequencing and data analysis

Fresh leaves of *Cha. luteum* were finely chopped using a razor blade in 1 ml of nuclei isolation buffer (0.25 M sucrose, 10 mM Tris-HCl pH 8.0, 10 mM MgCl₂, 1% Trion X-100, 5 mM β-mercaptoethanol) supplemented with 1× Halt™ Protease Inhibitor Cocktail (Thermo Scientific), and the slurry was filtered through a 50-μm cell strainer. The resulting nuclei were washed twice and resuspended with the same nuclei isolation buffer. An aliquot was analyzed using a flow cytometer for quality control and quantification. Based on the quantification, a volume containing approximately 75,000 nuclei was aliquoted, and the nuclei pellet was collected by centrifugation.

The nuclei pellet was resuspended in a transposition reaction mix containing Tagment DNA Enzyme (TDE1, Illumina, 20034197), 0.4× PBS, 0.01% digitonin, and 0.1% Tween-20, and incubated at 37 °C for 30 min. Transposition products were purified using the MinElute PCR Purification Kit (QIAGEN, 28004). Libraries were amplified with the NEBNext® High-Fidelity 2× PCR Master Mix (NEB, M0541), and further purified using VAHTS™ DNA Clean Beads (Vazyme, N411). The final libraries were sequenced in paired-end mode (2× 151 cycles) on the Illumina NovaSeq 6000 (Illumina Inc., USA).

Sequencing reads were adapter-trimmed using fastp (v0.20.0)[106] and aligned to the reference genome with BWA-MEM (v0.7.17)[107]. SAMtools (v1.16.1)[108] was used to remove duplicate and multi-mapping reads (-q 30), followed by peak calling (-q 0.01) with MACS (v3.0.0)[103]. For visualization, BAM files containing uniquely mapped reads were converted to BigWig format using deepTools bamCoverage (v3.5.1)[104], with Reads Per Kilobase per Million mapped reads (RPKM) normalization.

## Preparation of fluorescence in situ hybridization (FISH) probes

The consensus sequences of satellite repeats reconstructed by TAREAN (TAndem REpeat ANalyzer)[109] were used to design fluorescence-modified oligonucleotides which were synthesized by Eurofins (Germany). The clone pAtT4[110] was used as the probe to detect *Arabidopsis*-type telomeres. Plasmid DNA was labeled with ATTO550-dUTP using Fluorescent Nick Translation Labeling kits (Jena Bioscience, Germany).

The chromosome 2-specific oligo painting probes were designed based on the genome assembly of *Chi. japonica*[31] using the software Chorus2[111]. The predicted *Chionographis*-based oligos which matched the genome of *Cha. luteum* were all included in the synthesized myTags Immortal Libraries (Daicel Arbor Biosciences, USA) to improve the probe transferability to *Cha. lutuem*. Oligo pools were labeled with fluorophores ATTO-594, Alexa 488, or Alexa 647 following the myTags Immortal Labeling Protocol (Daicel Arbor Biosciences, USA, https://arborbiosci.com/wp-content/uploads/2022/05/DaicelArborBio_myTags_Labeling_Protocol_v2-2.pdf).

## Chromosome preparation and FISH

Mitotic chromosome spreads were prepared from root meristems using a dropping method[31]. Roots were pretreated in ice-cold water overnight, fixed in 3:1 (ethanol: glacial acetic acid) fixative at RT, overnight and kept in 70% ethanol at -20 °C until use. Fixed roots were digested in an enzyme mixture (0.7 % cellulose Onozuka R10 (Duchefa Biochemie, The Netherlands, cat. no. C8001), 0.7 % Cellulase (Calbiochem, USA, cat. no. 219466), and 1.0 % pectolyase (Sigma, USA, cat. no. 45-P3026)) in citric buffer (0.01 M sodium citrate dihydrate and 0.01 M citric acid) at 37 °C for 30–40 min. Cell suspension in the 3:1 fixative was dropped onto slides on a hot plate at 55 °C, and slides were further fixed in 3:1 fixative for 1 min, air-dried, and kept at 4 °C for later use.

To prepare meiotic chromosomes, inflorescences of *Chi. japonica* and *Cha. luteum* were fixed as described above for roots. Anthers were digested at 37 °C for 70-80 min in an enzyme mixture (0.23 % cellulose Onozuka R10 (Duchefa Biochemie, The Netherlands, cat. no. C8001), 0.23 % Cellulase (Calbiochem, USA, cat. no. 219466), 0.33 % pectolyase (Sigma, USA, cat. no. 45-P3026), and 0.33 % cytohelicase (Sigma, USA,

cat. no. C8247)). Meiotic spreads were prepared by a typical squash method[112]. FISH mapping was performed as described in ref. 35. For oligo- FISH, the hybridization mixture containing 10% dextran sulfate, 50% formamide, 2× SSC, and 500–1000 ng of each labeled oligo pool was used, and the hybridization at 37 °C was extended to 36–48 h.

## Microscopy and image analysis

Widefield fluorescence images were captured using an epifluorescence microscope BX61 (Olympus) equipped with a CCD camera (Orca ER, Hamamatsu, Japan) and pseudo-colored by the Adobe Photoshop 6.0 software. To analyze the chromatin and centromere ultra-structures at the super-resolution level, we applied spatial structured illumination microscopy (3D-SIM) using a 63 × /1.40 Oil Plan-Apochromat objective of an Elyra 7 microscope system (Carl Zeiss GmbH, Germany). Image stacks were captured separately for each fluorochrome using 405, 488, and 561 nm laser lines for excitation and appropriate emission filters[113]. Maximum intensity projections from image stacks were calculated using the ZENBlack software (Carl Zeiss GmbH, Germany). Zoom-in sections were presented as single slices to indicate the subnuclear chromatin structures at the super-resolution level. 3D rendering to produce spatial animations was done based on 3D-SIM image stacks using the Imaris 9.7 (Bitplane, UK) software. The tool 'Colocalization' of the same software was applied to determine the CENH3 amount colocalizing to the *Chama* repeats based on voxel intensities. For it, the colocalization calculation threshold was automatically determined at $P = 1.000$. Based on this, the percentage of the CENH3 volume above the threshold colocalizing to the *Chama* volume was calculated per interphase nucleus.

## Transmission electron microscopy (TEM)

For electron microscopic analysis, root tip cuttings were used for microwave-assisted fixation in 2.0% (v/v) glutaraldehyde and 2.0% (v/v) paraformaldehyde in 0.05 M cacodylate buffer (pH 7.3), dehydration with acetone, and embedding in Spurr's resin. Ultra-thin sections (70 nm) were cut with a Leica microtome Ultracut S (Leica Micro-systems, Wetzlar, Germany), and mounted on 70 mesh copper TEM grids. Prior to ultrastructure analysis at 120 kV in a Tecnai Sphera G2 transmission electron microscope (ThermoFisher Scientific, Eindho-ven, Netherlands), sections were contrasted in a Leica EM AC 20 automatic contrasting device with homemade 2% uranyl acetate for 30 min, followed by a 90-second incubation in Leica Ultrastain 2 con-taining 3% Reynolds' lead citrate.

## Statistics and reproducibility

No statistical method was used to predetermine the sample size, no data were excluded from the analyses, the experiments were not ran-domized, and the investigators were not blinded to allocation during the experiments and during outcome assessment.

## Reporting summary

Further information on research design is available in the Nature Portfolio Reporting Summary linked to this article.

## Data availability

The whole-genome sequencing and RNA-seq datasets generated for this study can be found at EMBL-ENA under the project IDs PRJEB82607 and PRJEB82608, respectively. The datasets of CENH3-ChIP-seq (Project ID: PRJNA1201173, accession GSE285103, https://www.ncbi.nlm.nih.gov/geo/query/acc.cgi?acc=GSE285103), histone mark ChIP-seq (ArrayExpress accession E-MTAB-16192, https://www.ebi.ac.uk/biostudies/arrayexpress/studies/E-MTAB-16192), and ATAC-seq (Project ID: PRJNA1201177, accession GSE285102, https://www.ncbi.nlm.nih.gov/geo/query/acc.cgi?acc=GSE285102) were deposited in the NCBI GEO or EMBL-EBI Annotare database. The gene annotation, syntenic genes, and sequences of oligo-FISH painting probes and proteins are available in Zenodo (https://zenodo.org/records/15182433). Source data are provided with this paper.

## Code availability

The Python scripts for genome and synteny analyses are available in BitBuckket (https://bitbucket.org/ipk-csf/chamaelirium2chiographis/).

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

## Acknowledgements

The authors thank Oda Weiss, Ines Walde, Manuela Knauft, Susanne König, and Pascal Jaroschinsky (IPK) for technical help in DNA isolation and sequencing, Anne Fiebig (IPK) for sequence submission, Kirsten Hoffie and Marion Benecke (IPK) for technical support in sample preparation for transmission electron microscopy, and Takayoshi Ishii (Arid Land Research Center, Tottori University, Japan) for providing plant materials; Klaus Mayer (Helmholtz Center Munich, Munich, Germany), André Marques (MPI Breeding Research, Cologne, Germany) and Martin Mascher (IPK) for suggestion on genome assembly; Ingo Schubert (IPK) and Noriyuki Tanaka (Tokyo, Japan) for discussion; and Neysan Donnelly for editorial suggestions. We thank the IPK for providing the technical infrastructure. This work was supported by the Deutsche Forschungsgemeinschaft DFG grants No. HO1779/32-1 and HO1779/32-2 (to A.Ho.); the Taiwan Ministry of Science and Technology (MOST) grants MOST 106-2313-B-002-034-MY3 and MOST 108-2811-B-002-608) (to Y.T.K.); the Czech Science Foundation grant 25-15809S (to P.N.); the China Scholarship Council CSC202006850005 (to J.C.); the scholarships from National Council for Scientific and Technical Research (CONICET) and the National Agency for the Promotion of Science and Technology (FONCyT) project PICT 2020–1777 (to M.A.S.).

## Author contributions

Y.T.K. performed the majority of the experiments, sequence and image analyses, including DNA and RNA isolation, immunostaining, FISH, ChIP-seq; P.N. and J.M. performed kinetochore protein and genome sequence analyses; J.C. performed genome assembly, gene annotation, and synteny analysis; V.S. performed super-resolution microscopy and image analysis; K.K. performed immunostaining; J.F. conducted flow cytometry; M.M. performed electron microscopy; M.A.S. performed FISH and data analysis; Z.Z. performed ATAC-seq and data analysis; A.Hi. performed PacBio and Hi-C library preparation and sequencing, H.H. grew and provided plant materials; A.Ho. supervised the research project; Y.T.K. and A.Ho. wrote the manuscript with the input from all coauthors.

## Funding

## Competing interests

The authors declare no competing interests.
