## [Transparent Peer Review file · Nature Communications]

Kinetochore mutations and histone phosphorylation pattern changes accompany holo- and macro-monocentromere evolution

Corresponding Author: Professor Andreas Houben

Version 0:

Reviewer comments:

Reviewer #1

(Remarks to the Author)

In this very interesting study, Kuo and co-workers report a fundamentally different centromere/kinetochore organization from the monocentric versus holocentric organizations of known centromeres. By comparing the sequence organization of the holocentric flowering plant *Chionographis japonica* genome with that of its close relative, *Chamaelirium luteum*, they discovered a radically different “macro-monocentromere” sequence organization, which corresponded to the unique cytological appearance of *C. luteum* centromeres. These are dominated by extensive contiguous satellite sequence heterochromatin and lack a primary constriction. By comparison of the amino acid sequences of CENH3, they observed rapid evolution of the N-terminal tail, and a search for kinetochore proteins showed that *Cha. luteum* showed differences for some highly conserved components. Consideration of the functions of these kinetochore proteins led to a satisfying stepwise model for how this unique macro-monocentromere organization diverged from a monocentric common ancestor. They argue that holocentricity arose strictly in *Chionographis*, while *Chamaelirium* underwent a localized centromere expansion. This striking discovery will be of considerable interest to the broad readership of Nature Communications.

The authors state that the chimeric origin of NSL1 and Borealin, which evidently triggered the chain of events, is unprecedented, but I wonder if there are other examples for species that have been sequenced but which haven't been studied cytologically with sufficient resolution to distinguish holocentricity from macro-monocentricity. For example, Lepidopterans are holocentric, exceptionally species rich, lack CENP-A/cenH3 and have small chromosomes, and perhaps there are clues in the sequences of their kinetochore proteins waiting to be discovered.

Minor:

Page 10, line 296, Is the KNL2 gene in *Chionographis* known to be functional? It seems better to say it is intact.

Page 7, lines 210-211 and Figure 2G: The text states that *Chamaelirium* has “the proportionally largest centromeres reported to date” and the figure displays the size ratio of centromere to chromosome as a percentage, with *Chamaelirium* centromeres encompassing 15.6 % of an average chromosome. The authors can consider whether to add *Solenopsis invicta*, in which the primary constriction and its CenSol satellite account for an average of 34% of the chromosome length [Huang et al 2016]. CenH3 localization has not been done, so it is possible that *Solenopsis* centromeres could be metapolycentric, like in *Tribolium castaneum*, where the distance between the most peripheral of the cenH3 dots accounts for 43% of the chromosome length [Grzan et al. 2020], or *Pisum* and *Lathyrus*, where the authors found the metapolycentric primary constriction encompasses a third of the chromosome.

(Remarks on code availability)

Reviewer #2

(Remarks to the Author)

This manuscript by Kuo et al is valuable not only for its novel research results but also for its thoughtful summaries of widely

different centromere and kinetochore structures, both at the chromosomal level and individual protein level. The figures are elegant, and few people can match the quality of their microscopy. In terms of novel results, the more important one here is *Chamaelirium luteus*, the sole member of the *Chamaelirium* genus and the most closely related species to the holocentric *Chionographis* genus, is monocentric but has unusual centromeres. They overlap large tandem arrays of a 60bp repeat, spanning up to 10 Mb total, and they do not form primary constrictions. Perhaps related, *Chamaelirium* and *Chionographis* have chromosomal distributions of histone phosphorylation in mitosis that is typical of holocentric species even though *Chamaelirium* is monocentric. Likewise, both have an unusual structural variant of the Borealin protein which is a core component of the chromosomal passenger complex. In addition, *Chamaelirium* but not *Chionographis* lacks the highly conserved protein KNL2, which normally functions in CENH3 loading to centromeres. These are important results that will be of interest to a large audience of scientists interested in centromeres, kinetochores, and chromosome biology.

I have two major concerns, however, with the manuscript in its current state. First, the Kuo et al conclude based on CENH3 ChIP-seq that the size of the *Chamaelirium* centromeres match the sizes of the tandem repeat arrays. However, the data appears to suggest that the functional centromere defined by CENH3 occupies only a small segment of these large arrays. If this is true, the centromeres may in fact be well within the normal range of centromere sizes, and *Chamaelirium* is not the outlier that Kuo et al have concluded it is. Second, they draw conclusions about order of events and about causation without clearly explaining the underlying logic. Both concerns are explained in more detail below, as well as other comments and suggestions.

Since the lack of a primary constriction in *Chamaelirium luteus* is one of the major results of the manuscript a few words on the significance of primary constrictions and how they form would be helpful.

Why are there only eight CENH3-interacting Chama arrays in the 22 scaffolds if there are 12 chromosomes?

The statement "The sequence arrays are highly homogeneous and consistent in their sequence orientation" needs further quantitative support. Supplementary Fig. 2 only shows an anecdotal example of a 5-kb region of a single Chama array.

The CENH3 ChIP-seq analysis included both multimapping and unique mapping reads, which will overestimate the size of the CENH3 occupied region if it overlaps with tandem repeats. For example, imagine that most of the CENH3 is concentrated in a single 1-Mb region of the array. When reads from those regions are mapped to the genome, they will be randomly distributed across the entire array.

The merely 2-fold enrichment of Chama for CENH3 suggests most of the Chama repeats are not bound to CENH3. This also suggests the actual centromere, as defined by CENH3 occupancy, is a small, possibly fluid, region within the large Chama array.

Related to the above two points, how complex is the sequence makeup of the centromeric Chama arrays? Are they interspersed with retrotransposons? Could there be enough polymorphism in them that uniquely mapping reads could estimate the actual size of the centromeres within the larger arrays?

The statement "ATAC-seq revealed that the entire *Cha. luteum* chromosome, except for the centromere, exists in an open chromatin state" is misleading. ATAC-seq reveals nucleosome free regions. Even within genes there are peaks and valleys of ATAC-seq reads. What this data reveals is that accessible chromatin regions are evenly distributed across the chromosomes except in centromeres.

While it is clear from these results that *Chamaelirium* centromeres are not syntenic with *Chionographis*, it is not clear whether *Chamaelirium* centromeres are syntenic with other monocentric relatives. Answering this question would increase the impact of this manuscript.

The logic behind the title of the paper and the evolution model expounded on figure 7 seems either severely lacking or needs to be communicated more clearly. It seems that every conclusion could be in the opposite direction. For example, could one also conclude from these data that constriction-free chromosomes arose first, which subsequently led to selection for a new Borealin variant? Or that formation of a holocentromere is what led to spread of novel repeats? Or that *Chamaelirium* monocentric centromeres reverted from *Chionographis*-like holocentromeres? Or that increased sister chromatid cohesion resulted in a need for stronger/more microtubule centromere attachments? Why do the data support the conclusions the authors make rather than these alternatives?

Figure 2D: What are the units of the Y axis?

Figure 3B. An X-axis scale would be helpful

Figure 3C. "CjSat5" is not explained in the legend or main text. What is it and why is it included in the figure?

(Remarks on code availability)

Reviewer #3

(Remarks to the Author)

This study provides a comprehensive comparison of two related species that have unusual centromere structures. The study was initiated to test if the unusually large centromeres of one species (*Chamaeliurium luteum*) might be a precursor to the evolution of holocentromeres in the other species (*Chionographis japonica*). The contributions here are mostly on the side of *luteum* since *japonica* had been the subject of a previous report. Notable observations are the large size of the satellite arrays in *luteum*, the fact that it has H3S10/S28 phosphorylation all along chromosome arms, that borealin is chimeric in both species, that NSL1 is chimeric in *luteum*, and that NDC80 and KNL1 are not observable by immunolocalization in *luteum*. The authors argue that changes in kinetochore proteins and phosphorylation led to changes in centromere structure. The strengths of the study are the state of the art genome characterization and superb microscopy. A weakness is that the work is entirely comparative, that no firm conclusions can be made about cause and effect.

Major comments:

1) Title. The word "preceded" should be removed from the title. An appropriate alternative would be "are correlated with". It is just as likely that changes in centromere structure favored the changes in kinetochore proteins and phosphorylation. Only the borealin mutation can be said to have preceded the divergence of these species, and the proposed connection of borealin to changes to centromere structure involves long series of speculations that have not been tested in any way.

2) Line 133-136. There is no evidence here that that H3 phosphorylation is "required but not sufficient" for holocentromere formation. This type of language is usually reserved for experiments where mutants are available or some type of overexpression is carried out. This study only involves staining and comparing.

3) Lines 209-211. The language here should be softened. Centromere size is defined by the area occupied by CENH3. Because the satellite array is homogenized, it is not possible to know where CENH3 is in the array by ChIP-seq. The microscopy that supports this is excellent but does not have the resolution to show that CENH3 is continuous across the array.

4) Lines 326-332 and 517-518. Neither KNL1 nor NDC80 could be detected by immunolocalization in *luteum*. This is perhaps the most surprising result in the study. If the KMN complex is absent that would suggest a completely different kinetochore structure. However, the as it is, the data are unconvincing. Indeed, both genes are present and expressed, so the data as shown suggest the KMN complex has a different form (or other proteins present) that change the accessibility of antibodies to their targets. If the authors want to suggest that KNL1 and NDC80 are absent, they need to do some western blots.

5) Line 383. Change from "in short, the divergent.." to "in short, we propose that the divergent..".

6) The discussion can be improved. I found it to be rambling and overly speculative. In one section the authors suggest that KNL2 mutation is the most common cause of major shifts in centromere structure, yet in the next section, a case is made for borealin as the instigating event in this clade. If both are involved, then, taken together, the data suggest that there are multiple kinetochore alterations over long time frames, and seem more consistent with the view that centromeres changed first and the kinetochore mutations followed.

(Remarks on code availability)

Reviewer #4

(Remarks to the Author)

This study focuses on a comparative analysis of two closely related species within the tribe Chionographideae that exhibit distinct centromere types. By integrating genomic, epigenomic, centromere structural, and kinetochore protein data, the research aims to elucidate the mechanisms underlying the transition from monocentric to holocentric chromosomes. The topic is novel, and the research framework is comprehensive, especially in its incorporation of multi-dimensional data across molecular, cytological, and structural levels, offering significant insights into evolutionary biology. However, concerns remain regarding the quality of the genome assemblies, the depth of functional annotation, and the completeness of the evidence supporting certain key conclusions. These aspects require further clarification or supplementation.

A key concern is that the genome assembly used for *Cha. luteum* in this study is only 749 Mb, representing about 84% of the estimated genome size (~887 Mb), with a contig N50 of 2.1 Mb. This is relatively low given current standards in the era of HiFi and Hi-C sequencing, especially for analyzing highly repetitive regions such as centromeres or detecting gene family loss/fusion structural variations, where misinterpretation is likely. Improving the genome assembly quality and performing full-length transcriptome sequencing for *Cha. luteum* would enhance the reliability of protein structure annotation and help verify gene models.

The formation mechanism of the chimeric genes Borealin and NSL1 remains unclear, and the supporting evidence is insufficient. The authors suggest that the N-terminal regions of these two chimeric genes originate from RPS17 and BORCS6, respectively, based mainly on sequence homology analysis. However, the following critical evidence is lacking.

No transcriptional validation of the fusion junctions (e.g., RT-PCR); No demonstration of adjacent structural changes on the chromosome (is this a true gene fusion or a misannotation?)

The conclusion that the holocentromeres in *Chionographis* arose de novo may be premature. The authors base this inference on the lack of synteny between the monocentromere position in *Cha. luteum* and the holocentromeric units in *Chi. japonica*, yet holocentromere formation is known to involve neocentromerization at multiple non-centromeric regions. The authors should consider conducting ancestral state reconstruction to infer whether the common ancestor of *Cha. luteum* and *Chi. japonica* was more likely monocentric or holocentric. Further searches for conserved unit positions and residual Chama-like centromeric sequences in *Chi. japonica* would also be valuable.

(Remarks on code availability)

Version 1:

Reviewer comments:

Reviewer #1

(Remarks to the Author)

The authors' responses to the reviewer comments are satisfactory. The re-written Discussion is an improvement.

Line 513: KNL should be KNL2

Lines 547-552 and lines 554-560 have duplicated phrases that appear to be an editing error.

Lines 548-549: Fix grammar: "the results could also be due to suggests antibody incompatibility".

Line 550: "contextor" should be "context or"

(Remarks on code availability)

Reviewer #2

(Remarks to the Author)

My comments have been addressed. I have three additional and very minor ones:

The title of Supplementary Table 4, "Percentage of CENH3 amount colocalizing to Chama repeats" is incorrect. It should be something like "Percentage of Chama repeats colocalizing with CENH3."

Detailed methods describing how numbers in Supplementary Table 4 were generated are missing.

While it is implicit in the methods that multimapping ChIP-seq reads are included in the analyses, it would be better to make this explicit since it is a non-standard method. It would be helpful to add a sentence such as this: "No filtering to remove multimapping reads was applied, so all analyses include both uniquely mapping."

(Remarks on code availability)

Reviewer #3

(Remarks to the Author)

The manuscript is much improved and all of my comments have been addressed. The discussion is now more organized, cautious and balanced, as is warranted.

The authors may want to look at lines 547-558, where the wording is duplicated and somewhat confusing.

Also, I could not locate the ChIP-seq or ATAC seq data at NCBI.

(Remarks on code availability)

I am not capable of reviewing code.

Reviewer #4

(Remarks to the Author)

Thank you for your detailed response and for addressing my questions. I have no further comments.

(Remarks on code availability)

Point-by-point response to reviewer's comments

Reviewer #1 (Remarks to the Author)

In this very interesting study, Kuo and co-workers report a fundamentally different centromere/kinetochore organization from the monocentric versus holocentric organizations of known centromeres. By comparing the sequence organization of the holocentric flowering plant *Chionographis japonica* genome with that of its close relative, *Chamaelirium luteum*, they discovered a radically different “macro-monocentromere” sequence organization, which corresponded to the unique cytological appearance of *C. luteum* centromeres. These are dominated by extensive contiguous satellite sequence heterochromatin and lack a primary constriction. By comparison of the amino acid sequences of CENH3, they observed rapid evolution of the N-terminal tail, and a search for kinetochore proteins showed that *Cha. luteum* showed differences for some highly conserved components. Consideration of the functions of these kinetochore proteins led to a satisfying stepwise model for how this unique macro-monocentromere organization diverged from a monocentric common ancestor. They argue that holocentricity arose strictly in *Chionographis*, while *Chamaelirium* underwent a localized centromere expansion. This striking discovery will be of considerable interest to the broad readership of Nature Communications.

The authors state that the chimeric origin of NSL1 and Borealin, which evidently triggered the chain of events, is unprecedented, but I wonder if there are other examples for species that have been sequenced but which haven't been studied cytologically with sufficient resolution to distinguish holocentricity from macro-monocentricity. For example, Lepidopterans are holocentric, exceptionally species rich, lack CENP-A/cenH3 and have small chromosomes, and perhaps there are clues in the sequences of their kinetochore proteins waiting to be discovered.

RESPONSE

Many thanks for your insightful comment. Macro-monocentromere possessing species might likely coexist in otherwise holocentric lineages. For example, a recent cytological finding in *Drosera* genus revealed that a macro-monocentromere-like structure in *D. filiformis* and holocentromeres in the related species *D. rotundifolia* (Hoshi and Terasaki 2025). However, no comparative kinetochore analysis has been performed for this genus. Future analysis of kinetochore genes in *Drosera* species could tell whether chimeric kinetochore genes were formed

and would provide additional insights into the interplay between kinetochore genes and centromere evolution.

We are not aware whether a similar situation exists in Lepidopterans. In moths and butterflies (in total ~160,000 species), only about 1.5 % of species have been karyotyped (De Vos et al. 2020), and only a small fraction have been analyzed in detail beyond simple chromosome counts. The key will be a detailed analysis of different species using a combination of methods, from microscopy to gene identification.

Minor:

Page10, line 296, Is the KNL2 gene in *Chionographis* known to be functional? It seems better to say it is intact.

RESPONSE

Thanks for the suggestion. We agree and change the “functional” to “intact”.

Now it reads (Page 10, Line 305-308): This *KNL2* fragment in *Cha. luteum* was found in the locus orthologous to the genomic region containing the intact *KNL2* gene in *Chi. japonica* (Supplementary Fig. 6B), indicating that it is a remnant of a formerly intact *KNL2* gene.

Page 7, lines 210-211 and Figure 2G: The text states that *Chamaelinium* has “the proportionally largest centromeres reported to date” and the figure displays the size ratio of centromere to chromosome as a percentage, with *Chamaelinium* centromeres encompassing 15.6 % of an average chromosome. The authors can consider whether to add *Solenopsis invicta*, in which the primary constriction and its CenSol satellite account for an average of 34% of the chromosome length [Huang et al 2016]. CenH3 localization has not been done, so it is possible that *Solenopsis* centromeres could be metapolycentric, like in *Tribolium castaneum*, where the distance between the most peripheral of the cenH3 dots accounts for 43% of the chromosome length [Grzan et al. 2020], or *Pisum* and *Lathyrus*, where the authors found the metapolycentric primary constriction encompasses a third of the chromosome.

RESPONSE

Thank you for pointing out additional cases of exceptionally large centromeric regions in *Solenopsis invicta*, *Tribolium castaneum*, *Pisum sativum*, and *Lathyrus*. We agree that these are highly relevant examples. We toned down our conclusion about the exceptional size of the active centromere (interacting with CENH3-positive nucleosomes) in *Cha. luteum*. Therefore, we removed Figure 2G and the corresponding Supplementary Table 4, but kept Figure 2H (now Figure 2G). This figure demonstrates directly the striking CENH3-size differences between *Cha. luteus* and a species (*Sorghum bicolor*) with similarly sized chromosomes.

However, in our original Figure 2G, we restricted the comparison to the CENH3-ChIPseq-defined centromeres in monocentric species. Thus, the metapolycentric species, which contain multiple centromere units within a defined chromosomal region on each chromosome and species in which centromere size estimates are based on cytological constrictions or satellite repeat abundance rather than direct CENH3 mapping, were not included. For example, in the well-studied metapolycentric chromosome 6 of *Pisum sativum*, the primary constriction (CEN6) spans 81.6 Mb (Macas et al. 2023), yet the total size of CENH3-enriched domains within this region is only 4.6 Mb, leaving a large region devoid of CENH3. While larger primary constrictions or extended centromeric regions have been described in some metapolycentric taxa (e.g., *Pisum* and *Lathyrus*) or in species where CENH3 localization has not yet been mapped (e.g., *Solenopsis*), our analysis was restricted to centromeres directly defined by CENH3-ChIP-seq in monocentric species.

Reviewer #2 (Remarks to the Author)

This manuscript by Kuo et al is valuable not only for its novel research results but also for its thoughtful summaries of widely different centromere and kinetochore structures, both at the chromosomal level and individual protein level. The figures are elegant, and few people can match the quality of their microscopy. In terms of novel results, the more important one here is *Chamaelirium luteus*, the sole member of the *Chamaelirium* genus and the most closely related species to the holocentric *Chionographis* genus, is monocentric but has unusual centromeres. They overlap large tandem arrays of a 60bp repeat, spanning up to 10 Mb total, and they do not form primary constrictions. Perhaps related, *Chamaelirium* and *Chionographis* have chromosomal distributions of histone phosphorylation in mitosis that is typical of holocentric

species even though *Chamaelirium* is monocentric. Likewise, both have an unusual structural variant of the Borealin protein which is a core component of the chromosomal passenger complex. In addition, *Chamaelirium* but not *Chionographis* lacks the highly conserved protein KNL2, which normally functions in CENH3 loading to centromeres. These are important results that will be of interest to a large audience of scientists interested in centromeres, kinetochores, and chromosome biology.

I have two major concerns, however, with the manuscript in its current state. First, the Kuo et al conclude based on CENH3 ChIP-seq that the size of the *Chamaelirium* centromeres match the sizes of the tandem repeat arrays. However, the data appears to suggest that the functional centromere defined by CENH3 occupies only a small segment of these large arrays. If this is true, the centromeres may in fact be well within the normal range of centromere sizes, and *Chamaelirium* is not the outlier that Kuo et al have concluded it is.

RESPONSE

We toned down our conclusion about the exceptional size of the active centromere (interacting with CENH3-positive nucleosomes) in *Cha. luteum*. Therefore, we removed Figure 2G and the corresponding Supplementary Table 4, but kept Figure 2H (now Figure 2G). This figure demonstrates directly the striking CENH3-size differences between *Cha. luteus* and a species (*Sorghum bicolor*) with similarly sized chromosomes.

Since our genome assembly of *Cha. luteum* is not complete; we quantified the *Chama* repeat proportion interacting with CENH3-containing nucleosomes by super-resolution microscopy. The new analysis demonstrates that ~60 % of *Chama* arrays interact with CENH3-containing nucleosomes (new Supplementary Table 4). We added to the text the following (Page 7, Line 197-202): “To determine the CENH3-positive fraction of the *Chama* arrays, we harnessed super-resolution microscopy. Colocalization of *Chama* FISH and CENH3 immunostaining interphase signals demonstrated that ~60 % of *Chama* arrays interact with CENH3-containing nucleosomes (new Supplementary Table 4). Thus, a substantial fraction of the macro-monocentromeric regions forms the active centromere.”

Second, they draw conclusions about order of events and about causation without clearly explaining the underlying logic. Both concerns are explained in more detail below, as well as other comments and suggestions.

RESPONSE

We rewrote the DISCUSSION and included description of the proposed model.

Since the lack of a primary constriction in *Chamaelirium luteus* is one of the major results of the manuscript a few words on the significance of primary constrictions and how they form would be helpful.

RESPONSE

The absence of a primary constriction in *Cha. luteus* demonstrates that a chromosome constriction is not a prerequisite for a functional monocentromere. Thus, the absence of a primary constriction does not always mean that no monocentromere exists. We assume that a direct or indirect interplay between the formation of a primary constriction and a pericentromere-specific phosphorylation of histone H3S10/S28 exist. A number of years ago, we (Houben et al. 1999) and others demonstrated that centromere inactivation correlates with the absence of pericentromeric histone H3 phosphorylation marks. In *Cha. luteum*, due to the chromosome-wide distribution of this cell cycle-dependent phosphorylation mark, no primary constriction is likely formed. We extended the DISCUSSION with (Page 19-20, Line 594-601): “The existence of constriction-free centromeres in *Cha. luteum* suggests that the primary constriction, while typically distinguishing monocentromeres, is not always necessary for centromere function. Consequently, the absence of this constriction does not necessarily indicate the absence of an active monocentromere. This morphological exception may be explained by the species' immense centromere size and the distribution of histone H3 phosphorylation marks along the entire length of the chromosomes; a pattern otherwise characteristic of holocentromeres.

Why are there only eight CENH3-interacting Chama arrays in the 22 scaffolds if there are 12 chromosomes?

RESPONSE

Thank you very much for this question. The lower number of CENH3-interacting *Chama* arrays than expected is caused by the challenges of assembling highly repetitive regions. Collapsing of sequence contigs occurs during genome assembly when repetitive DNA regions are incorrectly merged into a single, shorter contig, losing the true number of repeats and creating an inaccurate representation of the genome. This phenomenon is particularly prevalent with repetitive sequences, such as tandemly arrayed repeats, like the 60-bp-monomer *Chama* repeat which represents 9.3 % of the genome.

Our approach to assembling the *Cha. luteum* genome largely mirrored our successful strategy for *Chi. japonica* (Kuo et al. 2023a), where we generated a chromosome-scale reference genome despite a high proportion of centromeric repeats. However, *Cha. luteum* presented challenges much earlier in the process, beginning with the initial step of high-quality DNA isolation. Both *Cha. luteum* and *Chi. japonica* leaf tissues are rich in secondary metabolites that severely compromise DNA yield and quality. To circumvent this, we aimed to propagate a single plant via tissue culture and isolate sufficient high-molecular-weight (HMW) DNA from the roots; unfortunately, this initial step already proved difficult.

However, unfortunately, *Cha. luteum* grows extremely slowly and was not suitable for tissue culture in our hands. Therefore, we isolated genomic DNA from leaves from a pool of plants to obtain sufficient material for PacBio long-read sequencing, generating 101.7 Gb of PacBio CCS reads (~114×). However, the initial assembly was highly fragmented (14,573 contigs, N50 of ~652 kb). Moreover, the total assembled size was ~1.6 Gb, nearly twice the haploid genome size estimated by flow cytometry (1C = 887.4 Mb), which probably indicates an issue caused by pooling the plant materials for sequencing.

Genome assembly assessment of *Cha. luteum*

	First assembly	Second assembly	Assembly after purge_dups
No. of contigs	14,573	4,087	649
Largest contig	6,926,063	19,839,389	19,839,389
Total length	1,597,791,599	1,556,940,134	744,551,200
N50	625,023	1,631,558	2,111,717
N75	57,449	663,828	1,228,881
L50	564	192	87

L75	3163	564	205
GC (%)	41.55	40.51	41.13
Complete BUSCOs	-	2911 (90.0%)	2768 (85.5%)
Complete and single-copy BUSCOs	-	1167 (36.1%)	2477 (76.5%)
Complete and duplicated BUSCOs	-	1744 (53.9%)	291 (9.0%)
Fragmented BUSCO	-	171 (5.3%)	215 (6.6%)
Missing BUSCO	-	154 (4.7%)	235 (7.9%)
Total BUSCO groups searched	-	3236	3236

To overcome this, next, we extracted HMW DNA from the leaves of a single plant of *Cha. luteum* for a second PacBio run (53 Gb, ~59.5× coverage). Applying the same assembly approach, the improved assembly reduced contigs to 4,087 and increased the largest contig size (19.84 Mb), and raised the N50 to 1.63 Mb. Although we used Hifiasm (Cheng et al. 2021), a haplotype-resolved *de novo* assembler based on phased assembly graphs, the assembled genome size (~1.56 Gb) was nearly twice the 1C genome size of 887.4 Mb estimated by flow cytometry. This unexpectedly large assembly size, along with the high proportion of duplicated BUSCOs (68.0%), indicated substantial heterozygosity in the primary contigs. To address this, we applied `purge_dups` to remove redundant haplotypic sequences, which may also have included some centromeric repeat arrays. The refined assembly contained 649 contigs (N50 = 2.1 Mb, L50 = 87) and retained 92.7% complete BUSCOs. Finally, Hi-C scaffolding generated the top 22 large scaffolds, each exceeding 10 Mb (10–68 Mb), used for downstream analyses. The limited plant material from a single plant prevented us from performing Nanopore ultra-long sequencing to improve the assembly further.

Importantly, FISH mapping of the *Chama* repeat combined with CENH3 immunostaining revealed colocalization signals on all 12 chromosomes of *Cha. luteum*. This demonstrates that each chromosome harbors a *Chama*-associated centromere repeat array, even though not all are represented in the assembly. Due to the large and highly repetitive nature of the centromeres, the high heterozygosity of this genome, we didn't aim for a telomere-to-telomere level assembly in this study. Instead, our focus was on identifying and characterizing centromere-associated repeats, and we acknowledge that future improvements in sequencing and assembly strategies may resolve all CENH3-interacting *Chama* arrays at the chromosomal level.

The statement “The sequence arrays are highly homogeneous and consistent in their sequence orientation” needs further quantitative support. Supplementary Fig. 2 only shows an anecdotal example of a 5-kb region of a single *Chama* array.

RESPONSE

To support the statement “the sequence arrays are highly homogeneous and consistent in their sequence orientation”, we generated scaffold-wide and megabase-scale dotplots by Gepard (Krumstiek et al. 2007), with the *Chama* array-containing scaffold Clu_scaffold_17 as an example. We performed the analysis using two parameters: a less stringent word size of 20 (default = 10) to capture broader sequence similarity, and a strict word size of 60, matching the 60-bp *Chama* monomer length, to detect individual repeat units precisely. We include the result in the Supplementary Fig. 2D. Dotplot analysis under the relaxed condition (word size 20) showed that *Chama* arrays are highly homogeneous (Supplementary Fig. 2D, left). Under the strict condition, the analysis revealed the presence of higher-order repeat structure (Supplementary Fig. 2D, middle). Furthermore, *Chama* arrays exhibit a consistent orientation spanning regions larger than 1 Mb (Supplementary Fig. 2D, right), and many arrays maintain the same orientation across several hundred kilobases. In summary, these large-scale dotplot analyses confirm that *Chama* arrays are both highly homogeneous and consistently oriented, supporting the original statement. We added to the text (Page 7, Line 210-214): Unlike the frequently alternating orientation of the centromeric *Chio* satellite arrays observed in *Chi. japonica*³¹, in *Cha. luteum* the sequence arrays are highly homogeneous and consistently oriented, with higher-order repeat structures maintained across several hundred kilobases and extending in some cases over more than 1 Mb (Supplementary Fig. 2B–D).

The CENH3 ChIP-seq analysis included both multimapping and unique mapping reads, which will overestimate the size of the CENH3 occupied region if it overlaps with tandem repeats. For example, imagine that most of the CENH3 is concentrated in a single 1-Mb region of the array. When reads from those regions are mapped to the genome, they will be randomly distributed across the entire array.

The merely 2-fold enrichment of *Chama* for CENH3 suggests most of the *Chama* repeats are not bound to CENH3. This also suggests the actual centromere, as defined by CENH3 occupancy, is a small, possibly fluid, region within the large *Chama* array.

Related to the above two points, how complex is the sequence makeup of the centromeric *Chama* arrays? Are they interspersed with retrotransposons? Could there be enough polymorphism in them that uniquely mapping reads could estimate the actual size of the centromeres within the larger arrays?

RESPONSE

We agree with your concern. Analysis of the identified *Chama* repeat-containing centromere arrays did not reveal intermingling retrotransposons. Further, using ChIPseq-Mapper, a genome assembly-independent CENH3-ChIPseq analysis tool did not identify retrotransposons. Besides the *Chama* repeat, we identified two additional CENH3-enriched repeat clusters: the (AG/CT)_n-containing CL51 and GC-rich CL176 repeat clusters (Fig. 2C), both with very low genome abundance (<0.3%) and no sequence similarity to the *Chama* satellite. Neither of these is a retrotransposon-annotated repetitive sequence.

Apart from the *Chama* satellite, the remaining repeat clusters in the *Cha. luteum* genome exhibits a relatively low genomic proportion (<1.2%). Additionally, we performed FISH mapping of the two CRM LTR retrotransposons, CL66 and CL79, with genome proportions of 0.21 % and 0.18 %, respectively. None of them showed centromere-specific signals. Thus, we did not identify a CENH3 nucleosome-interacting retrotransposon in *Cha. luteum*. A similar satellite-dominant centromere composition was found in *Chi. japonica* (Kuo et al. 2023).

To determine the proportion of *Chama* array interacting with CENH3-containing nucleosomes, we employed a ChIP/genome assembly-independent method. Using super-resolution microscopy of anti-CENH3/*Chama*-labelled nuclei, we found that ~60 % of *Chama* arrays interact with CENH3-containing nucleosomes (new Supplementary Table 4). We added to the text (Page 7, Line 197-202): “To determine the CENH3-positive fraction of the *Chama* arrays, we harnessed super-resolution microscopy. Colocalization of *Chama* FISH and CENH3 immunostaining interphase signals demonstrated that ~60 % of *Chama* arrays interact with CENH3-containing nucleosomes (new Supplementary Table 4). Thus, a substantial fraction of the macro-monocentromeric regions forms the active centromere.”

The statement “ATAC-seq revealed that the entire *Cha. luteum* chromosome, except for the centromere, exists in an open chromatin state” is misleading. ATAC-seq reveals nucleosome free regions. Even within genes there are peaks and valleys of ATAC-seq reads. What this data reveals is that accessible chromatin regions are evenly distributed across the chromosomes except in centromeres.

RESPONSE

We rephrased the sentence. Now it reads (Page 8, Line 248-250): “In addition, ATAC-seq revealed that accessible chromatin regions are evenly distributed across the chromosomes except in the centromeres of *Cha. luteum* (Fig. 3B).

While it is clear from these results that *Chamaelirium* centromeres are not syntenic with *Chionographis*, it is not clear whether *Chamaelirium* centromeres are syntenic with other monocentric relatives. Answering this question would increase the impact of this manuscript.

RESPONSE

We thank the reviewer for this suggestion. Unfortunately, current genomic resources do not allow a direct test of centromere synteny with close relatives of *Cha. luteum*. The recent study by Henderson et al. (DOI: 10.21203/rs-7008504/v1) includes several monocot species, but none of them are representative of Melanthiaceae or Liliales, and all diverged from Liliales over 100 million years ago. By contrast, *Chamaelirium* and *Chionographis* diverged within *Melanthiaceae* only 20–40 million years ago. We also evaluated the recently published Melanthiaceae genomes by Zeng et al. (Nature Plants, DOI: 10.1038/s41477-025-02060-3). But these provided no centromere-specific annotation, and the data are insufficient for comparative analysis. Therefore, synteny with closer monocentric relatives cannot be assessed with the available data.

Moreover, the basic chromosome numbers (x) among the analyzed close monocentric relatives show significant variation: $x=17$ in *Heloniadeae* species, $x=8$ in *Veratrum*, and $x=5$ in *Paris* (Pellicer et al. 2014). Since this differs substantially from the $x=12$ found in *Chamaelirium* and *Chionographis*, such a high degree of karyotype divergence indicates that centromere synteny analysis across these species would likely lack relevance or meaning."

The logic behind the title of the paper and the evolution model expounded on figure 7 seems either severely lacking or needs to be communicated more clearly. It seems that every conclusion could be in the opposite direction. For example, could one also conclude from these data that constriction-free chromosomes arose first, which subsequently led to selection for a new *Borealin* variant? Or that formation of a holocentromere is what led to spread of novel repeats? Or that *Chamaelirium* monocentric centromeres reverted from *Chionographis*-like holocentromeres? Or that increased sister chromatid cohesion resulted in a need for stronger/more microtubule centromere attachments? Why do the data support the conclusions the authors make rather than these alternatives?

RESPONSE

We toned down the title, which now reads: “Kinetochores mutations and histone phosphorylation pattern changes accompany holo- and macro-monocentromere evolution” and extended the description of the evolution model and adjusted Figure 7.

Question: “could one also conclude from these data that constriction-free chromosomes arose first, which subsequently led to selection for a new *Borealin* variant?”

RESPONSE

We agree with your suggestion. Now it reads (Page 14, Line 434-438): “The change in the *Borealin* gene might have acted as a molecular trigger, initiating the chromosome-wide distribution of histone phosphorylation and driving the evolution of primary constriction-free chromosomes in *Chionographideae*. Conversely, constriction-free chromosomes may have arisen first, followed by the selection for the new *Borealin* variant”.

Question: “Or that formation of a holocentromere is what led to spread of novel repeats?”

RESPONSE

The mechanism(s) that result(s) in the formation of a holocentromere are unknown. However, it is likely that partial similarity exists between *de novo* formation process of monocentromeres and

holocentromeres. However, in the case of a neo-monocentromere, the newly formed centromere often forms in the proximity of the “older” centromere. In contrast, the mono-to-holocentromere transition process is accompanied by the formation of many *de novo* formed centromere units along all chromosomes. Whether the formation of a metapolycentromere (e.g. *Lathyrus*, *Pisum*) is involved in this process as intermediate has been debated but not proven yet (Schubert et al. 2020). In metapolycentric *Lathyrus sativus*, all centromere domains are composed of the same satellite DNA family. However, in metapolycentric *Pisum* species closely related to *Lathyrus*, even centromere units on the same chromosome are based on distinct repeats. Thus, it is not clear how metapolycentromeres evolved. Thus, multiple conceivable scenarios, including the one mentioned by the reviewer.

It's plausible that the size of the centromere unit increased gradually in *Chi. japonica*. Under this model, a small-unit holocentromere first formed *de novo* (by an unknown mechanism). Over time, these small units would have undergone massive expansion, reaching sizes of several megabases (Mb) through repetitive DNA accumulation facilitated by mechanisms such as unequal crossing over and gene conversion.

Question: “Or that Chamaelirium monocentric centromeres reverted from Chionographis-like holocentromeres?”

RESPONSE

A *de novo* origin of macro-monocentromeres from holocentromeres is improbable for two main reasons. First, this process would necessitate multiple fissions of the holocentric chromosomes, which should result in a large, unobserved increase in the number of monocentric chromosomes. Alternatively, it could occur through the elimination of most holocentromere units from the genome, preceded or followed by changes in CENH3 localization. Such "reversion scenario" cannot be completely ruled out, but is unlikely because a mono-to-holo-to-mono transition is more complex than a mono-to-holo transition. Second, there are no documented cases of an evolutionary reversion from a holocentric state back to monocentricity. We added (Page 14, Line 411-415): “ A *de novo* origin of macro-monocentromeres from holocentromeres is unlikely for two main reasons. First, such a process would require multiple fissions of holocentric chromosomes, resulting in a

large number of monocentric chromosomes, which is not observed. Second, there are no documented cases of a reversion from holocentricity to monocentricity.”

Question: “Or that increased sister chromatid cohesion resulted in a need for stronger/more microtubule centromere attachments?”

RESPONSE

We agree with this scenario. We have written (Page 20, Line 603-611): “Accurate chromosome segregation relies on a delicate balance between sister chromatid cohesion and microtubule-generated pulling forces (Fig. 7B). Consequently, as an adaptation required for proper chromosome segregation, the expansion of centromeric regions in both *Chamaelirium* and *Chionographis* probably increased microtubule attachment and pulling forces to offset the increased sister chromatid cohesion brought on by the chromosome-wide distribution of pericentromeric histone H3 phosphorylation. This sequence of events is more likely than the alternative scenario of independent histone H3 phosphorylation expansions in both genera, followed by centromere expansion.”

Figure 2D: What are the units of the Y axis?

RESPONSE

The Y axis of the ChIP-seq track is the average log₂ ratio of IP/input reads in a genome-wide 10 kb windows. The Y axis of the gene track is the coverage calculated as the number of reads per 1-kb bin. We included the units in Figure 2D.

Figure 3B. An X-axis scale would be helpful

RESPONSE

We included the X-axis in the image.

Figure 3C. "CjSat5" is not explained in the legend or main text. What is it and why is it included in the figure?

RESPONSE

CjSat5, a satellite repeat identified in the *Chi. japonica* genome (Kuo et al. 2023b), was mapped onto chromosome 2. Our analysis demonstrated that high-copy CjSat5 sequences dominate the two distal ends of the chromosome. Because of this extreme enrichment, the design of oligo FISH-painting probes was restricted, and we could not extend them into these terminal regions. Now the legend of Figure 3C reads: (Page 45, Line 1494-1497) "Our mapping revealed that the ends of chromosome 2 are highly enriched with the CjSat5 satellite repeat, precluding the design of oligo FISH-painting probes for those specific regions."

Reviewer #3 (Remarks to the Author)

This study provides a comprehensive comparison of two related species that have unusual centromere structures. The study was initiated to test if the unusually large centromeres of one species (*Chamaeliurium luteum*) might be a precursor to the evolution of holocentromeres in the other species (*Chionographis japonica*). The contributions here are mostly on the side of *luteum* since *japonica* had been the subject of a previous report. Notable observations are the large size of the satellite arrays in *luteum*, the fact that it has H3S10/S28 phosphorylation all along chromosome arms, that borealin is chimeric in both species, that NSL1 is chimeric in *luteum*, and that NDC80 and KNL1 are not observable by immunolocalization in *luteum*. The authors argue that changes in kinetochore proteins and phosphorylation led to changes in centromere structure. The strengths of the study are the state of the art genome characterization and superb microscopy. A weakness is that the work is entirely comparative, that no firm conclusions can be made about cause and effect.

RESPONSE

We thank your thoughtful summary and highlighting both the strengths and limitations of our study. We agree that our work is primarily comparative and does not allow us to draw firm conclusions

about the direct cause and effect. This limitation stems largely from the uniqueness of the non-model plant materials, which constrain functional experimentation. We tested transgenic approaches, but we failed with this material so far. Nevertheless, we believe that the combination of genomic and cytological approaches provides an important foundation for future experimental work. We hope our findings will serve as a framework for testing direct causal relationships in kinetochore and centromere evolution as new experimental tools become available and for inspiring studies in other independently evolved lineages.

Major comments:

1) Title. The word “preceded” should be removed from the title. An appropriate alternative would be “are correlated with”. It is just as likely that changes in centromere structure favored the changes in kinetochore proteins and phosphorylation. Only the borealin mutation can be said to have preceded the divergence of these species, and the proposed connection of borealin to changes to centromere structure involves long series of speculations that have not been tested in any way.

RESPONSE

We thank you for your suggestion. We agree that “preceded” in the title could imply a cause-effect relationship, which has not been experimentally tested. We have revised the title to use “accompany”.

Now the revised title reads: “Kinetochore mutations and histone phosphorylation pattern changes accompany holo- and macro-monocentromere evolution.”

2) Line 133-136. There is no evidence here that H3 phosphorylation is “required but not sufficient” for holocentromere formation. This type of language is usually reserved for experiments where mutants are available or some type of overexpression is carried out. This study only involves staining and comparing.

RESPONSE

We thank you for pointing this out. We agree that using “required but not sufficient” could imply a functional test that cannot be performed in both species. To avoid an overstatement, we have revised the text.

Now it reads: (Page 5, Line 133-140) “Additionally, comparison with representatives of the closely related monocentric tribe Heloniadeae revealed that a distinct distribution pattern of the cell cycle-dependent histone H3 phosphorylation marks in the Tribe Chionographideae. Notably, the distribution of these modifications along the entire length of chromosomes does not distinguish between holo- and macro-monocentric species.”

3) Lines 209-211. The language here should be softened. Centromere size is defined by the area occupied by CENH3. Because the satellite array is homogenized, it is not possible to know where CENH3 is in the array by ChIP-seq. The microscopy that supports this is excellent but does not have the resolution to show that CENH3 is continuous across the array.

RESPONSE

We toned down our conclusion about the exceptional size of the active centromere (interacting with CENH3-positive nucleosomes) in *Cha. luteum*. Therefore, we removed Figure 2G, but kept Figure 2H (now Figure 2G). Figure 2G, demonstrates directly the striking CENH3-size differences between *Cha. luteum* and *Sorghum bicolor*, a species with similarly sized chromosomes.

Since our genome assembly of *Cha. luteum* is not complete; we quantified the *Chama* repeat proportion interacting with CENH3-containing nucleosomes by super-resolution microscopy. The new analysis demonstrates that ~60 % of *Chama* arrays interact with CENH3-containing nucleosomes (new Supplementary Table 4). We added to the text (Page 7, Line 197-202): “To determine the CENH3-positive fraction of the *Chama* arrays, we harnessed super-resolution microscopy. Colocalization of *Chama* FISH and CENH3 immunostaining interphase signals demonstrated that ~60 % of *Chama* arrays interact with CENH3-containing nucleosomes (new Supplementary Table 4). Thus, a substantial fraction of the macro-monocentromeric regions forms the active centromere.”

4) Lines 326-332 and 517-518. Neither KNL1 nor NDC80 could be detected by immunolocalization in luteum. This is perhaps the most surprising result in the study. If the KMN complex is absent that would suggest a completely different kinetochore structure. However, the as it is, the data are unconvincing. Indeed, both genes are present and expressed, so the data as shown suggest the KMN complex has a different form (or other proteins present) that change the accessibility of antibodies to their targets. If the authors want to suggest that KNL1 and NDC80 are absent, they need to do some western blots.

RESPONSE

We agree that the absence of detectable KNL1 and NDC80 signals in our immunolocalization experiments does not prove that these proteins are absent. Both genes are indeed present in the genome and expressed in transcriptome data. The lack of detectable signals most likely reflects antibody incompatibility, or conformational masking of epitopes in the kinetochore context, rather than the absence of the proteins themselves.

Unfortunately, due to the technical constraints, we are unable to perform Western blotting at this time. To avoid overinterpretation, we have revised the text to state that KNL1 and NDC80 were not detected by immunolocalization with the antibodies used in this study, and we refrain from concluding that they are absent. We also discuss possible reasons for the lack of immunodetection, as suggested by the reviewer. Now it reads (Page 18, Line 557-567): “NSL1, as a component of MIS12c, is one of the key structural kinetochore proteins. It is essential for interaction with the other two complexes of the outer kinetochore, NDC80c and KNL1c. In *Drosophila*, NSL1 knockout mutants are lethal, underscoring its essential role for kinetochore function. The effects of the fusion of NSL1 with the N-terminal region of BORCS6 on the formation of KMN complex in centromeres of *Cha. luteum* remain unclear. While the lack of KNL1 and NDC80 immunosignals could indicate dramatic changes in kinetochore composition the results could also be due to suggests antibody incompatibility or conformational masking of epitopes within the kinetochore context or inaccessibility of antibody target domain. Thus, further investigation is needed to clarify the kinetochore composition and function in *Cha. luteum*.”

5) Line 383. Change from “in short, the divergent..” to “in short, we propose that the divergent..”.

RESPONSE

Thank you very much for the suggestion, we rewrote the Discussion. This sentence was removed.

6) The discussion can be improved. I found it to be rambling and overly speculative. In one section the authors suggest that KNL2 mutation is the most common cause of major shifts in centromere structure, yet in the next section, a case is made for borealin as the instigating event in this clade. If both are involved, then, taken together, the data suggest that there are multiple kinetochore alterations over long time frames, and seem more consistent with the view that centromeres changed first and the kinetochore mutations followed.

RESPONSE

We rewrote most of the DISCUSSION and considered the option that the centromeres change first and the kinetochore mutations follow.

Reviewer #4 (Remarks to the Author)

This study focuses on a comparative analysis of two closely related species within the tribe Chionographideae that exhibit distinct centromere types. By integrating genomic, epigenomic, centromere structural, and kinetochore protein data, the research aims to elucidate the mechanisms underlying the transition from monocentric to holocentric chromosomes. The topic is novel, and the research framework is comprehensive, especially in its incorporation of multi-dimensional data across molecular, cytological, and structural levels, offering significant insights into evolutionary biology. However, concerns remain regarding the quality of the genome assemblies, the depth of functional annotation, and the completeness of the evidence supporting certain key conclusions. These aspects require further clarification or supplementation.

A key concern is that the genome assembly used for *Cha. luteum* in this study is only 749 Mb, representing about 84% of the estimated genome size (~887 Mb), with a contig N50 of 2.1 Mb. This is relatively low given current standards in the era of HiFi and Hi-C sequencing, especially

for analyzing highly repetitive regions such as centromeres or detecting gene family loss/fusion structural variations, where misinterpretation is likely. Improving the genome assembly quality and performing full-length transcriptome sequencing for *Cha. luteum* would enhance the reliability of protein structure annotation and help verify gene models.

RESPONSE

We agree with your concern about the genome assembly of *Cha. luteum*. Unlike expected, we faced several problems. Our approach to assembling the *Cha. luteum* genome largely mirrored our successful strategy for *Chi. japonica* (Kuo et al. 2023a), where we generated a chromosome-scale reference genome despite a high proportion of centromeric repeats. However, *Cha. luteum* presented challenges much earlier in the process, beginning with the initial step of high-quality DNA isolation. Both *Cha. luteum* and *Chi. japonica* leaf tissues are rich in secondary metabolites that severely compromise DNA yield and quality. To circumvent this, we aimed to propagate a single plant via tissue culture and isolate sufficient high-molecular-weight (HMW) DNA from the roots; unfortunately, this initial step already proved difficult.

However, unfortunately, *Cha. luteum* grows extremely slowly and was not suitable for tissue culture in our hands. Therefore, we isolated genomic DNA from leaves from a pool of plants to obtain sufficient material for PacBio long-read sequencing, generating 101.7 Gb of PacBio CCS reads (~114×). However, the initial assembly was highly fragmented (14,573 contigs, N50 of ~652 kb). Moreover, the total assembled size was ~1.6 Gb, nearly twice the haploid genome size estimated by flow cytometry (1C = 887.4 Mb), which probably indicates an issue caused by pooling the plant materials for sequencing.

Genome assembly assessment of *Cha. luteum*

	First assembly	Second assembly	Assembly after purge_dups
No. of contigs	14,573	4,087	649
Largest contig	6,926,063	19,839,389	19,839,389
Total length	1,597,791,599	1,556,940,134	744,551,200
N50	625,023	1,631,558	2,111,717
N75	57,449	663,828	1,228,881
L50	564	192	87
L75	3163	564	205
GC (%)	41.55	40.51	41.13

Complete BUSCOs	-	2911 (90.0%)	2768 (85.5%)
Complete and single-copy BUSCOs	-	1167 (36.1%)	2477 (76.5%)
Complete and duplicated BUSCOs	-	1744 (53.9%)	291 (9.0%)
Fragmented BUSCO	-	171 (5.3%)	215 (6.6%)
Missing BUSCO	-	154 (4.7%)	235 (7.9%)
Total BUSCO groups searched	-	3236	3236

To overcome this, next, we extracted HMW DNA from the leaves of a single plant of *Cha. luteum* for a second PacBio run (53 Gb, ~59.5× coverage). Applying the same assembly approach, the improved assembly reduced contigs to 4,087 and increased the largest contig size (19.84 Mb), and raised the N50 to 1.63 Mb. Although we used Hifiasm (Cheng et al. 2021), a haplotype-resolved *de novo* assembler based on phased assembly graphs, the assembled genome size (~1.56 Gb) was nearly twice the 1C genome size of 887.4 Mb estimated by flow cytometry. This unexpectedly large assembly size, along with the high proportion of duplicated BUSCOs (68.0%), indicated substantial heterozygosity in the primary contigs. To address this, we applied `purge_dups` to remove redundant haplotypic sequences, which may also have included some centromeric repeat arrays. The refined assembly contained 649 contigs (N50 = 2.1 Mb, L50 = 87) and retained 92.7% complete BUSCOs. Finally, Hi-C scaffolding generated the top 22 large scaffolds, each exceeding 10 Mb (10–68 Mb), used for downstream analyses. The limited plant material from a single plant prevented us from performing Nanopore ultra-long sequencing to improve the assembly further.

However, in our study, we did not intend to decipher the complete genome of *Cha. luteum*. We aimed for insight into the evolution of atypical centromeres accompanying the mono- to holocentric transition. As demonstrated by our comparative chromosome painting experiment (Figure 3C–E), we confirmed by FISH the accuracy of the obtained genome analysis, because the employed FISH probes were designed based on the determined sequences. Also, the obtained genome assembly in combination with ChIP-seq, immunostaining, and FISH was sufficient for the characterization of the macro-monocentromeres.

The formation mechanism of the chimeric genes *Borealin* and *NSL1* remains unclear, and the supporting evidence is insufficient. The authors suggest that the N-terminal regions of these two chimeric genes originate from *RPS17* and *BORCS6*, respectively, based mainly on sequence homology analysis. However, the following critical evidence is lacking. No transcriptional validation of the fusion junctions (e.g., RT-PCR); No demonstration of adjacent structural changes on the chromosome (is this a true gene fusion or a misannotation?)

RESPONSE

We understand your concern. We have now provided multiple lines of evidence to validate the chimeric nature of *Borealin* and *NSL1* genes at both the genomic and transcriptional levels:

- 1) Transcriptomic evidence: Transcripts corresponding to the two genes were also identified in a genome-independent transcriptome assembly (Trinity) derived from RNA-seq data, confirming that these chimeric sequences are not assembly artefacts.
- 2) Transcriptional validation: We performed RT-PCR for both chimeric *Borealin* and *NSL1* genes, targeting the fusion junctions. The amplified products match the expected sizes, and Sanger sequencing of these products confirms the chimeric origin. The gel image and sequencing results are now provided in Supplementary Fig. 8A-B, and the corresponding materials and methods are included in the main text (Page 25-26, line 774-787).
- 3) Genomic evidence: Chimeric *Borealin* (fused *RPS17* and *Borealin*) and *NSL1* (fused *BORCS6* and *NSL1*) are present in single PacBio reads from both *Cha. luteum* and *Chi. japonica*. These reads span the entire gene structure, including exons and introns, of both genes. We provide a new Supplementary Fig. 8C illustrating these PacBio reads and gene structures.

Taken together, these results provide strong evidence that *Borealin* and *NSL1* are genuine chimeric genes rather than misannotations or assembly artefacts. Now it reads in the main text (Page 11, line 326-328): "The chimeric nature of *Borealin* and *NSL1* genes was further confirmed by transcriptional validation and genomic evidence (Supplementary Fig. 8)."

The conclusion that the holocentromeres in *Chionographis* arose de novo may be premature. The authors base this inference on the lack of synteny between the monocentromere position in *Cha. luteum* and the holocentromeric units in *Chi. japonica*, yet holocentromere formation is known to involve neocentromerization at multiple non-centromeric regions. The authors should consider

conducting ancestral state reconstruction to infer whether the common ancestor of *Cha. luteum* and *Chi. japonica* was more likely monocentric or holocentric. Further searches for conserved unit positions and residual Chama-like centromeric sequences in *Chi. japonica* would also be valuable.

RESPONSE

Our conclusion regarding the *de novo* origin of holocentromeres in *Chionographis* is supported by the observed broad-scale synteny between its holocentric *Chi. japonica* and that of macro-monocentric *Cha. luteum*, with the notable exception of their centromeres. We found a high degree of genome conservation between both species, despite their divergence approximately 23.5 million years ago. Specifically, syntenic blocks in *Chi. japonica* were found to be split by centromere units, which we interpreted as an indication of the *de novo* origin of these centromere units. The positions of the monocentromeres in *Cha. luteum* and the centromere units in *Chi. japonica* do not always correspond according to syntenic genes, suggesting that the loss of monocentromeres in *Chionographis* was likely accompanied by *de novo* holocentromere formation. A *de novo* origin of macro-monocentromeres derived from holocentromeres is unlikely, because a *de novo* formation of monocentric chromosomes based on multiple fission events of a holocentric chromosome would result in numerous monocentric chromosomes. However, the chromosome number of *Cha. luteum* is in the range of *Chionographis* species. In addition, it is very likely that a common ancestor of *Cha. luteum* and *Chi. japonica* was monocentric because all known closely related species are monocentric (see Figure 1a, based on the phylogenetic trees of (Pellicer et al. 2014; Kim et al. 2019; Givnish et al. 2016). The tribes, holo/macro-monocentric Chionographideae and monocentric Heloniadeae, appear to have originated in North America, at 23.5 mya in the early Miocene and 27.3 mya in the late Oligocene, respectively, with the ancestors of *Chionographis* and monocentric *Heloniopsis* – monocentric *Ypsilandra* (China, Japan, and Korea) dispersing into East Asia. Thus, both tribes show an intercontinental disjunction between East Asia and North America at the intergeneric level.

Our findings are consistent with the reviewer's point that holocentromere formation is known to involve neocentromerization at multiple non-centromeric regions. Our study revealed that the *de novo* holocentromere formation sites in *Chi. japonica* correspond to gene-rich, accessible chromatin regions in the syntenic *Cha. luteum* genome. This observation directly aligns with the concept of neocentromerization occurring in regions that were previously non-centromeric and transcriptionally active, rather than at pre-existing centromeric loci.

While the comparative repeat analysis using RepeatExplorer did not reveal shared high-copy repeats between *Heloniopsis umbellata* and either *Cha. luteum* or *Chi. japonica*, we did identify a specific conserved 9-base pair sequence (TTCGTACGA) that is shared between the *Chama* monomer in *Cha. luteum* and the *Chio* monomer in *Chi. japonica*. Suggesting a potential ancient shared feature or functional significance.

Question: “Further searches for conserved unit positions and residual Chama-like centromeric sequences in *Chi. japonica* would also be valuable.”

RESPONSE

Since we could not identify conserved unit position, we could not identify residual Chama-like centromeric sequences in *Chi. japonica*. Also, after removing the shared 9-bp conserved motif between *Chio* and *Chama* monomers, the remaining *Chama* sequence showed no significant similarity to the genome of *Chi. japonica* in BLASTn (v2.5.0, default parameters). Given the absence of conserved centromeric sequence features and extensive karyotype divergence among closely related taxa (e.g. Heloniadeae species with n=17, compared with n=12 in *Chamaelirium* and *Chionographis*), an ancestral state reconstruction based solely on current genomic and cytological data would not yield a reliable inference. Therefore, while we can't definitely exclude alternative evolutionary scenarios, the available evidences best supports the interpretation that holocentromeres in *Chionographis* arose *de novo*, rather than through modification of ancestral monocentromeres.

References

Cheng H, Concepcion GT, Feng X, Zhang H, Li H (2021) Haplotype-resolved de novo assembly using phased assembly graphs with hifiasm. *Nature methods* 18 (2):170-175. doi:10.1038/s41592-020-01056-5

- De Vos J, Augustijnen H, Bättscher L, Lucek K (2020) Speciation through chromosomal fusion and fission in Lepidoptera. *Philosophical Transactions of the Royal Society B* 375:20190539. doi:<https://doi.org/10.1098/rstb.2019.0539>
- Givnish TJ, Zuluaga A, Marques I, Lam VKY, Gomez MS, Iles WJD, Ames M, Spalink D, Moeller JR, Briggs BG, Lyon SP, Stevenson DW, Zomlefer W, Graham SW (2016) Phylogenomics and historical biogeography of the monocot order Liliales: out of Australia and through Antarctica. *Cladistics* 32 (6):581-605. doi:10.1111/cla.12153
- Hoshi Y, Terasaki K (2025) The AT-rich regions correspond to the localized centromeres in meiotic chromosomes of *Drosera filiformis* Raf. *Cytologia* 90 (2):101-108. doi:10.1508/cytologia.90.101
- Houben A, Wako T, Furushima-Shimogawara R, Presting G, Kunzel G, Schubert I, Fukui K (1999) The cell cycle dependent phosphorylation of histone H3 is correlated with the condensation of plant mitotic chromosomes. *Plant Journal* 18 (6):675-679
- Kim C, Kim SC, Kim JH (2019) Historical biogeography of Melanthiaceae: A case of out-of-North America through the Bering land bridge. *Front Plant Sci* 10:396. doi:10.3389/fpls.2019.00396
- Kuo Y-T, Câmara AS, Schubert V, Neumann P, Macas J, Melzer M, Chen J, Fuchs J, Abel S, Klocke E, Huettel B, Himmelbach A, Demidov D, Dunemann F, Mascher M, Ishii T, Marques A, Houben A (2023) Holocentromeres can consist of merely a few megabase-sized satellite arrays. *Nat Commun* 14 (1):3502. doi:10.1038/s41467-023-38922-7
- Macas J, Avila Robledillo L, Kreplak J, Novak P, Koblizkova A, Vrbova I, Burstin J, Neumann P (2023) Assembly of the 81.6 Mb centromere of pea chromosome 6 elucidates the structure and evolution of metapolycentric chromosomes. *PLoS Genet* 19 (2):e1010633. doi:10.1371/journal.pgen.1010633
- Pellicer J, Kelly LJ, Leitch IJ, Zomlefer WB, Fay MF (2014) A universe of dwarfs and giants: genome size and chromosome evolution in the monocot family Melanthiaceae. *New Phytol* 201 (4):1484-1497. doi:10.1111/nph.12617
- Schubert V, Neumann P, Marques A, Heckmann S, Macas J, Pedrosa-Harand A, Schubert I, Jang TS, Houben A (2020) Super-resolution microscopy reveals diversity of plant centromere architecture. *Int J Mol Sci* 21 (10). doi:10.3390/ijms21103488

REVIEWERS' COMMENTS

Reviewer #1 (Remarks to the Author):

The authors' responses to the reviewer comments are satisfactory. The re-written Discussion is an improvement.

Line 513: KNL should be KNL2

RESPONSE

We have corrected KNL to KNL2.

Lines 547-552 and lines 554-560 have duplicated phrases that appear to be an editing error.

Lines 548-549: Fix grammar: "the results could also be due to suggests antibody incompatibility".

Line 550: "contextor" should be "context or"

RESPONSE

We have corrected the errors and deleted the duplicated lines 554-560. Now it reads (line 547-552): While the lack of KNL1 and NDC80 immunosignals could indicate dramatic changes in kinetochore composition, the results could also arise from antibody incompatibility, conformational masking of epitopes within the kinetochore context, or inaccessibility of antibody target domain. Thus, further investigation is needed to clarify the kinetochore composition and function in *Cha. luteum*.

Reviewer #2 (Remarks to the Author):

My comments have been addressed. I have three additional and very minor ones:

The title of Supplementary Table 4, "Percentage of CENH3 amount colocalizing to Chama repeats" is incorrect. It should be something like "Percentage of Chama repeats colocalizing with CENH3."

RESPONSE

We quantified the proportion of CENH3 signal volume that colocalizes with Chama repeat signals. Accordingly, we have revised the title of Supplementary Table 4 to: "Percentage of CENH3 signal colocalizing with *Chama* repeats measured in 16 *Cha. luteum* root interphase nuclei".

Detailed methods describing how numbers in Supplementary Table 4 were generated are missing.

RESPONSE

We have added the methods in the Microscopy and image analysis of the Methods part. Now it reads (page 32, line 987-993): 3D rendering to produce spatial animations was done based on 3D-SIM image stacks using the Imaris 9.7 (Bitplane, UK) software. The tool 'Colocalization' of the same software was applied to determine the CENH3 amount colocalizing to the *Chama* repeats based on voxel intensities. For it, the colocalization calculation threshold was automatically determined at $P=1.000$. Based on this, the percentage of the CENH3 volume above the threshold colocalizing to the *Chama* volume was calculated per interphase nucleus.

While it is implicit in the methods that multimapping ChIP-seq reads are included in the analyses, it would be better to make this explicit since it a non-standard method. It would be helpful to add a sentence such as this: "No filtering to remove multimapping reads was applied, so all analyses include both uniquely mapping."

RESPONSE

We agree with this suggestion. We have added “No filtering to remove multimapping reads was applied; therefore, all analyses include both uniquely and multimapping reads.” in the ChIP-seq data analysis of Methods part (page 29, line 900-901).

Reviewer #3 (Remarks to the Author):

The manuscript is much improved and all of my comments have been addressed. The discussion is now more organized, cautious and balanced, as is warranted.

The authors may want to look at lines 547-558, where the wording is duplicated and somewhat confusing.

RESPONSE

We have corrected the errors and deleted the duplicated lines 554-560. Now it reads (page 18, line 547-551): While the lack of KNL1 and NDC80 immunosignals could indicate dramatic changes in kinetochore composition, the results could also arise from antibody incompatibility, conformational masking of epitopes within the kinetochore context, or inaccessibility of antibody target domain. Thus, further investigation is needed to clarify the kinetochore composition and function in *Cha. luteum*.

Also, I could not locate the ChIP-seq or ATAC seq data at NCBI.

RESPONSE

We apologize for the inconvenience. Now, the CENH3-ChIP-seq and ATAC-seq data are accessible through the links.

CENH3-ChIP-seq (GEO accession GSE285103):

Access: <https://www.ncbi.nlm.nih.gov/geo/query/acc.cgi?acc=GSE285103>

ATAC-seq (GEO accession GSE285102)

Access: <https://www.ncbi.nlm.nih.gov/geo/query/acc.cgi?acc=GSE285102>

The newly included histone mark-ChIP-seq datasets have been submitted to the european GEO Annotare (accession number E-MTAB-16192) and are now in curation.

Reviewer #3 (Remarks on code availability):

I am not capable of reviewing code.

RESPONSE

The code used in this study was made publicly available, ensuring transparency and reproducibility.

Reviewer #4 (Remarks to the Author):

Thank you for your detailed response and for addressing my questions. I have no further comments.

RESPONSE

We thank you for the positive feedback and for confirming that all your concerns have been addressed.